# Wss1 metalloprotease partners with Cdc48/Doa1 in processing genotoxic SUMO conjugates

Maxim Y Balakirev[1,2,3]*, James E Mullally[4†], Adrien Favier[5], Nicole Assard[1,2,3], Eric Sulpice[1,2,3], David F Lindsey[6], Anastasia V Rulina[1,2,3], Xavier Gidrol[1,2,3], Keith D Wilkinson[4]*

[1]Institut de recherches en technologies et sciences pour le vivant-Biologie à Grande Echelle, Commissariat a l'Energie Atomique et aux Energies Alternatives (CEA), Grenoble, France; [2]Institut de recherches en technologies et sciences pour le vivant-Biologie à Grande Echelle, University Grenoble Alpes, Grenoble, France; [3]Biologie à Grande Echelle, Institut national de la santé et de la recherche médicale (INSERM), Grenoble, France; [4]Department of Biochemistry, Emory University, Atlanta, United States; [5]Institut de Biologie Structurale, University Grenoble Alpes, Grenoble, France; [6]Department of Biological Sciences, Walla Walla University, College Place, United States

*For correspondence: maxim.
balakirev@cea.fr (MYB);
genekdw@emory.edu (KDW)

Present address: †Office of In
vitro Diagnostic Devices and
Radiological Health, U.S. Food
and Drug Administration, Silver
Spring, United States

Competing interests: The
authors declare that no
competing interests exist.

Reviewing editor: Ivan Dikic,
Goethe University Medical
School, Germany

**Abstract** Sumoylation during genotoxic stress regulates the composition of DNA repair complexes. The yeast metalloprotease Wss1 clears chromatin-bound sumoylated proteins. Wss1 and its mammalian analog, DVC1/Spartan, belong to minigluzincins family of proteases. Wss1 proteolytic activity is regulated by a cysteine switch mechanism activated by chemical stress and/or DNA binding. Wss1 is required for cell survival following UV irradiation, the smt3-331 mutation and Camptothecin-induced formation of covalent topoisomerase 1 complexes (Top1cc). Wss1 forms a SUMO-specific ternary complex with the AAA ATPase Cdc48 and an adaptor, Doa1. Upon DNA damage Wss1/Cdc48/Doa1 is recruited to sumoylated targets and catalyzes SUMO chain extension through a newly recognized SUMO ligase activity. Activation of Wss1 results in metalloprotease self-cleavage and proteolysis of associated proteins. In cells lacking Tdp1, clearance of topoisomerase covalent complexes becomes SUMO and Wss1-dependent. Upon genotoxic stress, Wss1 is vacuolar, suggesting a link between genotoxic stress and autophagy involving the Doa1 adapter.

## Introduction

Maintenance of genome integrity is essential for cell survival and faithful transfer of genetic information. Eukaryotic cells evolved a DNA damage response (DDR) to sense, evaluate, and repair chromosome and DNA lesions or to direct cells for destruction when the harm is irreparable (*Ciccia and Elledge, 2010*). This dynamic process is under precise spatiotemporal control, requiring posttranslational modifications (PTM); phosphorylation, ubiquitylation, and sumoylation, to name a few (*Polo and Jackson, 2011*). Compared to phosphorylation, bulky modifications with ubiquitin (Ub), and SUMO provide additional scaffolding for regulation of composition, function, and stability of protein assemblies (*Jackson and Durocher, 2013*). Both modifiers are conjugated to target proteins by the concerted action of activating enzymes (E1), conjugating enzymes (E2) and ligases (E3) and both form chains through repeated cycles of self-conjugation (*Kerscher et al., 2006*). The effects of modification depend on its position on the target protein, chain length, and type of lysine (K) linkage.

**eLife digest** DNA repair is essential for cell survival. Every time DNA is damaged, several protein complexes sense the damage and act to repair it. These complexes need to be carefully regulated. One way this is achieved is by the addition of molecular tags that change the activity of these complexes. Sumoylation is one such modification, which involves the addition of a bulky molecular tag called SUMO.

Sumoylation during DNA damage is known to regulate the precise assembly and activity of the repair complexes. This modification is reversible and when the DNA repair is completed, the SUMO tags are removed and the repair complexes are disassembled. A protein called Cdc48 was known to work together with other molecules to clear SUMO-modified complexes from the DNA after the repair is complete. But it was unclear how this occurred and what roles other proteins played in the process.

Balakirev et al. now analyze the detailed workings of another protein called Wss1 and how it contributes to SUMO processing in yeast cells. The experiments show that Wss1 helps to remove the SUMO-modified complexes from the DNA by forming a complex with Cdc48 and the Cdc48-adaptor protein Doa1. Wss1 is a protease, an enzyme that can break down proteins, but it is inactive under the normal conditions inside a cell. Wss1 is found in the cell's nucleus (which contains most of the cell's DNA) until it senses DNA damage, which it does by recognizing damage-specific forms of DNA (such as single stranded DNA) and the SUMO tag.

Balakirev et al. found that Wss1 binds to the site of DNA damage and lengthens the SUMO tag. This indicates that Wss1 can also act as a ligase—an enzyme that helps to assemble polymeric SUMO. The polymeric SUMO in turn leads to the accumulation of more Wss1 and activates its protease activity. The protease cleaves the associated proteins in the repair complex, thus helping to extract the SUMO-modified proteins from the DNA. DNA damage also results in the transfer of Wss1 into a compartment inside the cell, called a vacuole. This suggests that autophagy—a mechanism used by cells to break down damaged cellular components—is one way that proteins are removed from the nucleus.

Together, Balakirev et al.'s findings reveal a previously unknown role for Wss1 and introduce us to another level of control in the DNA damage response. The next challenges will be to identify specific cellular components involved in transporting Wss1 to the vacuole and to examine whether this mechanism is conserved with a human version of Wss1, the Spartan/DVC1 protein.

---

Conjugation is reversible; specific proteases ensure regeneration of the modifier pool and regulate the PTM by reversing the modification, or remodeling the chains (*Eletr and Wilkinson, 2014*).

Ub and SUMO are involved in the displacement of proteins from chromatin (*Meyer et al., 2012*; *Jackson and Durocher, 2013*; *Vaz et al., 2013*). Assembly of repair complexes during genotoxic stress requires clearing the damaged and stalled chromatin components from the site of DNA damage. During or at the completion of repair, the assembled repair complexes must be disassembled. Ub-dependent protein degradation by the 26S proteasome is a major mechanism of protein clearing, involving Cdc48-promoted disassembly of chromatin-bound complexes (*Meyer et al., 2012*; *Vaz et al., 2013*). Cdc48, an AAA-ATPase chaperone long-thought to be Ub-specific, also interacts with sumoylated substrates via one of its multiple cofactors, Ufd1 (*Nie et al., 2012*). This interaction is required to protect cells from Topoisomerase (Top1)-induced DNA damage. The Cdc48/Ufd1/Npl4 complex is also targeted to sumoylated Rad52 and negatively regulates its interaction with Rad51, inhibiting excessive recombination (*Bergink et al., 2013*). Thus, Cdc48 plays a central role in DDR, acting as a Ub and SUMO-dependent segregase assembling and disassembling macromolecular complexes.

Other DDR components such as SUMO-targeted Ub ligases or STUbLs (*Sriramachandran and Dohmen, 2014*) exhibit dual specificity. STUbLs bind to sumoylated and poly-sumoylated proteins via SUMO-interacting motifs (SIMs) and catalyze their ubiquitylation. Sumoylated–ubiquitylated substrates may then be degraded by the proteasome. Cdc48 may also function parallel to STUbLs in another partially redundant pathway since SUMO binding by Cdc48 becomes essential in cells with STUbL mutations (*Nie et al., 2012*).

Wss1 may also be required for clearance of SUMO conjugates in the absence of functional STUbL (*Mullen et al., 2010, 2011*). Wss1, Weak Suppressor of *smt3-331*, is a suppressor of the temperature-sensitive (*ts*) phenotype produced by an L26S SUMO mutation (*Biggins et al., 2001*) and is implicated in the response to genotoxic stress through unknown mechanisms (*O'Neill, 2004*). Bioinformatics suggested that Wss1 belongs to a new family of metalloproteases (*Iyer et al., 2004*). Recent work demonstrated that it is a DNA-dependent protease involved in repair of DNA–protein crosslinks (*Stingele et al., 2014*). These authors also showed that Wss1 is important for cell survival upon topoisomerase 1 (Top1)-inflicted DNA damage. Notably, this function of Wss1 is assisted by Cdc48.

Here, we show that Wss1 is a SUMO-specific Cdc48 cofactor involved in clearing high molecular weight SUMO conjugates (HMW-SUMO). Important novel observations demonstrate that Wss1 binds SUMO and has both SUMO ligase and latent metalloprotease activities; is regulated by a cysteine switch mechanism that can be activated by chemical stress and/or DNA binding; forms a ternary complex with Cdc48 and Doa1 that exhibits specificity for SUMO and promotes binding of HMW-SUMO to Cdc48; is an inactive metalloprotease under normal conditions and promotes poly-sumoylation of proteins with which it interacts, including Cdc48; is accumulated within nuclear foci and its metalloprotease activity is activated upon genotoxic stress; is accumulated in the vacuole during stress, suggesting an intriguing link between SUMO-dependent DDR and vacuolar degradation. These studies confirm and extend those of Stingele et al. and suggest a unique model of environmental switching of Wss1 from a ligase to a protease in response to DNA damage.

## Results

### In vitro properties of WSS1
#### Wss1 is a SUMO-binding protein
Wss1 is predicted to be a metalloprotease, the prototype of the WLM superfamily of proteins (Wss1-like metalloproteases) (*Iyer et al., 2004*). Sequence alignment of Wss1 with 11 related proteins revealed highly conserved motifs, the most extensive being the metalloprotease WLM domain (*Figure 1A* and *Figure 1—figure supplement 1*). Consensus secondary structure prediction and 3D modeling support a metzincin-like fold for the WLM domain (*Gomis-Ruth, 2003*, *Figure 1—figure supplement 2*) most similar to minigluzincins (*Lopez-Pelegrin et al., 2013*), and the closest mammalian Wss1 analog, DVC1/Spartan (*Figure 1—figure supplement 2*). Additional motifs are two Cdc48-binding sequences (SHP and VIM), and a putative SUMO-interacting motif, SIM2 (*Figure 1A*). Another SIM, SIM1, is located just upstream of SIM2 and is also present in all 12 proteins. Finally, these modeling results suggest that the N-terminal region of Wss1 might adopt a beta-grasp fold characteristic of SUMO and Ub (data not shown).

Because of the predicted SIMs and its role in SUMO metabolism, we examined SUMO binding by Wss1. Hemagglutinin (HA)-tagged Wss1 and mutants were expressed in bacteria and their interaction with SUMO-beads was analyzed (*Figure 1B*). The beads bound HA-Wss1, suggesting a direct interaction with SUMO. Mutation of the conserved H and E residues of the WLM active site (WLM*) did not affect binding, but deletion of SIM2 (ΔSIM2), mutations in SIM1 (SIM1*), or both (SIM1*; ΔSIM2) resulted in a significant decrease in binding. Complete inhibition of Wss1 binding to SUMO was only achieved by triple mutation (SIM1*; ΔSIM2, WLM*) suggesting that all these elements are involved in the interaction with SUMO. This was further confirmed by the observation that both N-terminal (Wss1 1–188, containing WLM and SHP) and C-terminal (Wss1 189–269, bearing the VIM and SIMs) fragments of Wss1 were retrieved on SUMO beads. The interaction was specific because it was inhibited by free SUMO in solution and not by Ub.

Because Wss1 was reported to be Ub/SUMO isopeptidase (*Mullen et al., 2010*) and the WLM domain contributes to SUMO binding, we hypothesized that WLM might bind the carboxy-terminus of SUMO. To test this we synthesized a new ligand, SUMO-PA, where C-terminal Gly was replaced by amino-methyl-phosphonic acid ('Materials and methods'). Replacement of the C-terminal carboxyl by phosphonic acid should strengthen the coordination of the active site Zn, stabilizing a complex between metalloprotease and peptide substrate (*Mucha et al., 2010*). The SUMO-PA ligand was then linked to beads and used to measure Wss1 binding (*Figure 1C*). To further increase the selectivity, pull-downs were done in the presence of 1% Tx-100 to attenuate the hydrophobic SIM-SUMO interaction. Under these conditions no Wss1 was bound to SUMO, while SUMO-PA beads still retrieve

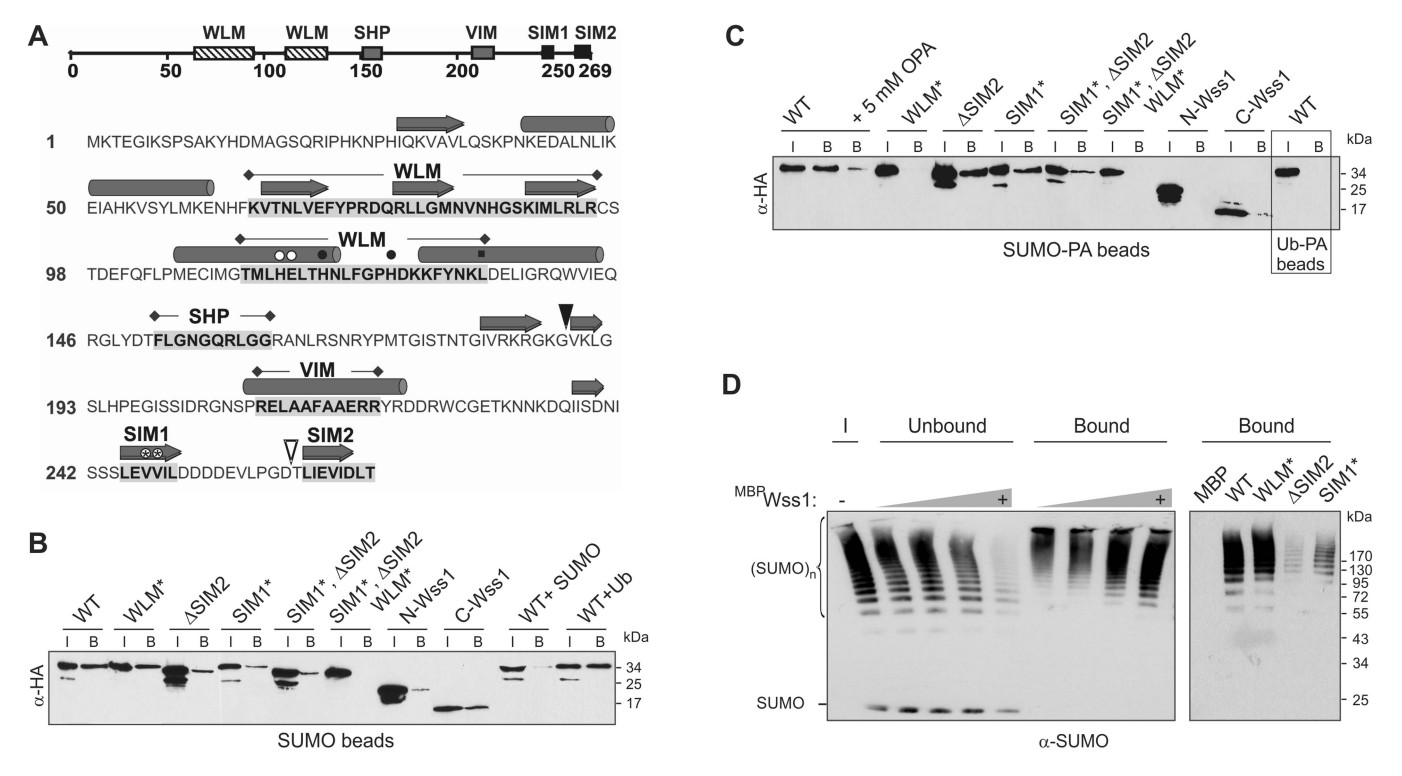

**Figure 1**. Wss1 is a SUMO-targeted metalloprotease. (**A**) Structure of Wss1 protein. Amino acids of the active site are marked by circles; structural Leu 133 of the active site is marked by square; empty circles show HE115,116NK mutation in WLM* construct; asterisks show VV247,248AA mutation in SIM1* construct; empty triangle shows truncation in ΔSIM2 construct; filled triangle shows Wss1 self-cleavage site that splits protein into N-Wss1 and C-Wss1. (**B**) Wss1 binding to SUMO beads. Recombinant HA-Wss1 proteins were produced with *pRSET-B* vector in bacteria and purified (see *Figure 1A* and *Supplementary file 4* for constructs description). SUMO beads were prepared by SUMO crosslinking to CNBr-activated Sepharose ('Materials and methods'). The 'input' (I) and 'bound' (B) fractions were analyzed by western blotting with α-HA. For competition 10 mg/ml of SUMO or Ub was used. Input: 20%. (**C**) Wss1 binding to SUMO-phosphonic acid (SUMO-PA) beads. SUMO-PA was synthesized by native chemical ligation and coupled to CNBr-activated Sepharose ('Materials and methods'). The binding assay was performed as in *Figure 1B* except that the concentration of Triton X-100 in binding buffer was ten times higher (1%). Addition of o-phenantroline is indicated as OPA. As a control experiment, the binding of Wss1 to Ub-PA beads is shown in the box. Input: 20%. (**D**) Wss1 affinity for SUMO chains. Recombined MBP-Wss1 proteins were expressed with *pMAL-c2* vector in bacteria and purified ('Materials and methods'). Their capacity to bind preformed SUMO chains was analyzed by western blotting with α-SUMO (all fractions were analyzed at the same dilution). Left panel: increasing amount of MBP-Wss1 WT protein was used to pull down SUMO chains with amylose beads. The 'input' (I), 'bound', and 'unbound' fractions are shown. Right panel: comparative efficiency of SUMO chain pull-down by MBP-Wss1 mutants bound to the same amount of amylose beads. Prior the experiment, the beads was loaded with an excess of the indicated MBP protein.

The following figure supplements are available for figure 1:

**Figure supplement 1**. Structural organization of WLM proteins.

**Figure supplement 2**. Structural comparison of WLM domain.

significant amount of the protein. Pointing to the role of Zn, the binding to SUMO-PA was inhibited by o-phenanthroline (OPA) and WLM* mutation, while ablation of SIMs had only a minor effect. Thus, the WLM domain appears to bind the C-terminus of SUMO and introduction of the phosphonic acid moiety enhances this interaction. The double SIM mutant showed diminished binding and the N-terminal-Wss1 fragment didn't bind SUMO-PA at all, indicating that SIM-SUMO interaction still contributed to the binding. On the other hand, the phosphonic acid moiety alone was not sufficient to insure binding since no Wss1 was retrieved with Ub-PA beads (*Figure 1C*).

The presence of multiple SIMs in Wss1 suggests that it might have higher affinity for polysumoylated substrates. Indeed amylose pull-down experiments with recombinant MBP-Wss1 demonstrated that

Wss1 preferentially binds HMW-SUMO chains (*Figure 1D*). Both SIM1* and ΔSIM2 mutation inhibit binding of SUMO chains, whereas mutation in WLM domain has no effect (*Figure 1D*).

## Wss1 metalloprotease is activated by ssDNA

Recombinant Wss1 lacks protease activity (*Stingele et al., 2014*), possibly due to improper protein folding or a specific cofactor requirement. Because we observed that GFP-Wss1 formed nuclear foci (see below) and others have reported activation by DNA (*Stingele et al., 2014*), we examined whether Wss1 protease activity can be triggered by nucleic acids and other negatively charged macromolecules. Recombinant Wss1 was purified under denaturing conditions and refolded in the presence of heparin, SDS, plasmid DNA, or ssDNA. Unexpectedly, we found that refolding in the presence of ssDNA-induced autocatalytic cleavage of the Wss1 protein (*Figure 2A*). Uncleaved Wss1 and one fragment (N-Wss1) precipitated along with the DNA while the other (C-Wss1) was soluble. Sequencing of the latter revealed cleavage at V189 (*Figure 2A*). No cleavage was observed with the WLM* mutant suggesting an autoproteolytic mechanism. As nuclease S1 inhibited Wss1 processing, the cleavage cofactor seems to be ssDNA polymer and not individual nucleotides.

To reduce precipitation of Wss1, the protein was first refolded and then a shorter 70 bp ssDNA was added (*Figure 2B*). Addition of ssDNA to Wss1 resulted in measurable protein cleavage after 3 hr of incubation. This reaction was inhibited by OPA, suggesting metalloprotease activation. Curiously, the cleavage pattern was different from that previously observed (*Figure 2A*) with multiple bands around 17 kD and negligible amount of large N-Wss1 fragment (*Figure 2B*). Thus, the pathways of Wss1 cleavage may differ depending on experimental conditions.

## Wss1 metalloprotease is activated by a cysteine-switch mechanism

During these experiments, we observed that the treatment of Wss1 with OPA induced protein oligomerization (*Figure 2B*). The oligomers are thiol-sensitive suggesting that they are disulfide linked (see *Figure 4D* below). We hypothesize that in Wss1 protein, the active site Zn is linked to a cysteine residue, which is released by OPA and promotes the formation of intermolecular crosslinks. This observation raises an interesting possibility that Wss1 is regulated by a cysteine-switch mechanism (*Chakraborti et al., 2003*). The metalloproteases regulated by a cysteine-switch mechanism can be activated by thiol-reacting reagents. By screening a panel of different electrophiles, we found that thiram (*Balakirev and Zimmer, 1998*), a compound that reversibly modifies cysteine residues, efficiently induces Wss1 self-cleavage (*Figure 2B*). Importantly, the cleavage pattern was the same as in refolding experiments (*Figure 2A*). Thiram belongs to the thiuram disulfide class of reactive compounds that modify protein sulfhydryl groups via thiol-disulfide exchange reaction (*Neims et al., 1966*; *Vallari and Pietruszko, 1982*) (*Figure 2—figure supplement 1*). Because thiram also inhibited OPA-induced Wss1 oligomerization (*Figure 2B*), it seems that it targets the same cysteine residue that is involved in Zn binding. In parallel, we found that toxic organomercury compound APMA, classically used for matrix metalloproteases activation, also produces Wss1 self-cleavage and inhibits OPA-dependent protein oligomerization (*Figure 2—figure supplement 2A,B*).

Together these data suggest that, like some other metalloproteases, Wss1 is regulated by a cysteine-switch mechanism. Displacement of the cysteine from the active site Zn induces autoproteolytic cleavage and may serve to release an inhibitory protein fragment. Wss1 has three cysteine residues, two of which, though located within the WLM domain, are far from the active site (*Figure 2—figure supplement 2C*). Moreover, their orientation suggests that they may form a disulfide bridge, though the estimated distance of 5.6 Å implies a conformational change would be necessary. The third cysteine, C226, is located in a negatively charged C-terminal part of the protein that was released by self-cleavage. This residue is conserved within the WLM family (*Figure 1—figure supplement 1*) and is a perfect candidate for the regulatory cysteine. Unfortunately, we were unable to produce soluble C226S protein, probably because of rapid auto-proteolysis of this construct. To examine the role of V189-cleavage site (*Figure 2A*) in C-terminal fragment release and activation of Wss1, we produced a Wss1 mutant where the GKG residues preceding V189 were replaced by AQA (*Figure 2—figure supplement 3*). Upon thiram treatment, the AQA mutant efficiently self-processed, though producing different proteolytic fragments (*Figure 2—figure supplement 3*). This result suggests that the primary sequence within this protein region is not essential for Wss1 activation.

To further address the role of the cysteine-switch mechanism in the DNA activation pathway, we examined the effect of different thiols on DNA-dependent Wss1 self-cleavage (*Figure 2D*). While thiram greatly accelerated this reaction, thiols, particularly 10 mM glutathione, significantly inhibited

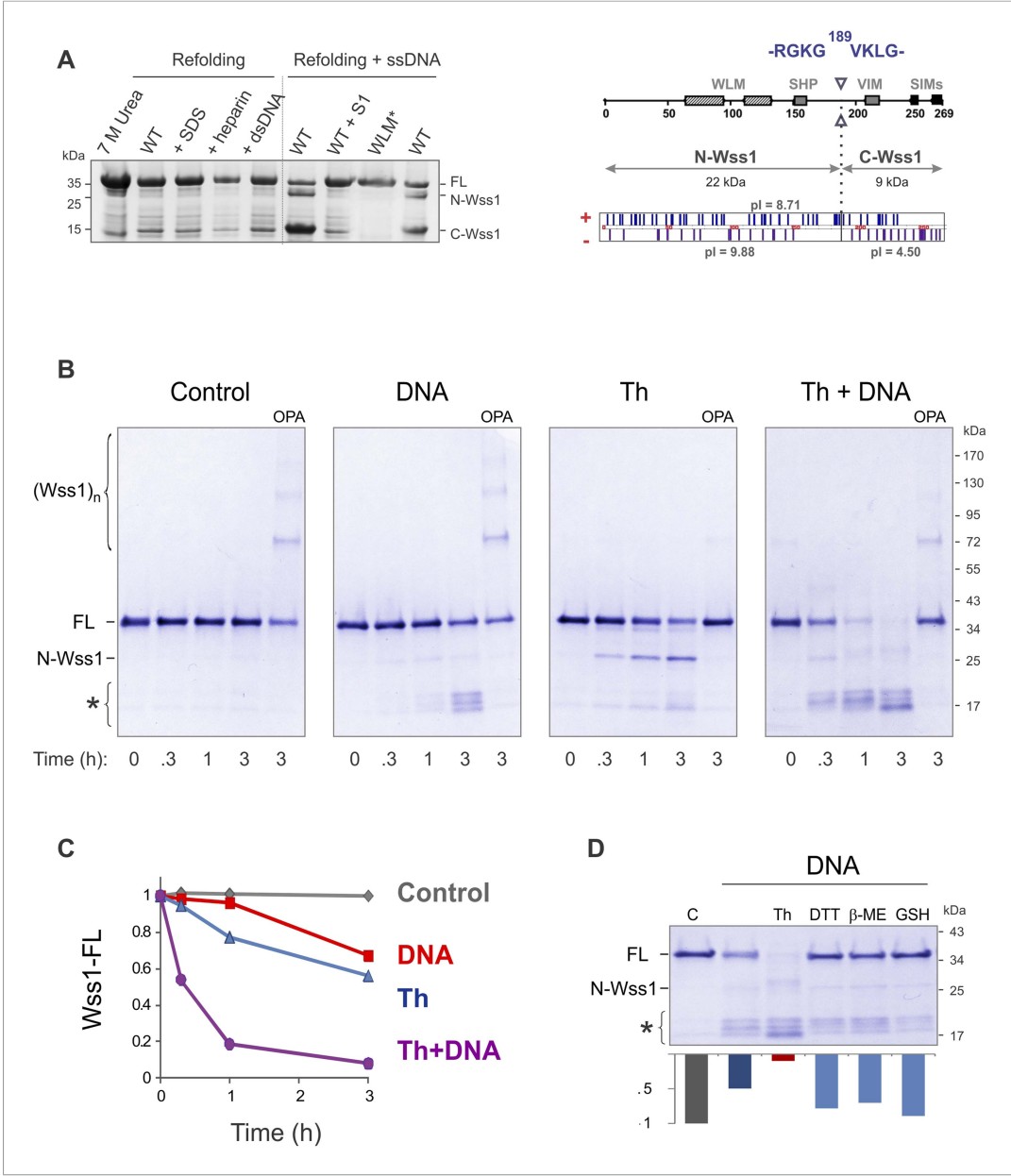

Figure 2. Protease activity of Wss1 protein. (A) Wss1 self-cleavage is induced by ssDNA. Wss1-His6 protein was purified by Ni-NTA chromatography in denaturing conditions and refolded by gradient dialysis in the presence of various cofactors, including SDS (0.1%), heparin (200 µg/ml), plasmid DNA (100 µg/ml), and ssDNA (M13mp18, 100 µg/ml). The samples were analyzed by PAGE with Coomassie staining. Refolding in the presence of ssDNA produced soluble (C-Wss1) and insoluble fragments (FL and N-Wss1). The reaction was inhibited by pre-treatment of ssDNA with S1 nuclease (5 U/ml, 30 min at 37˚C). No cleavage was observed with WLM* mutant. The diagram on the right shows the result of sequencing of soluble C-terminal fragment: site of cleavage, protein pI, and molecular weight as well as distribution of positively and negatively charged residues. (B) Activation of Wss1 cleavage by various conditions. Recombinant HA-Wss1 protein (200 µg/ml) was incubated for the indicated time at 25˚C. Where indicated 2.5 µM DNA (DNA, 70b mbpTop1d oligonucleotide), 0.5 mM thiram (Th), or 3 mM OPA was added at the beginning of the incubation. The reaction was analyzed by non-reducing gel electrophoresis and Coomassie staining and quantified by ImageJ. The positions of full-length HA-Wss1 (FL), large N-terminal fragment (N-Wss1), small cleavage products (asterisk), and HA-Wss1 oligomers (Wss1)n are shown. (C) Kinetics of Wss1 cleavage. Diagram shows the time course of the level of full-length HA-Wss1 at various conditions. The data are obtained by ImageJ quantification of the experiments shown on *Figure 2B*. (D) Effect of thiol reagents on DNA-dependent Wss1 activation. Recombinant HA-Wss1 protein (200 µg/ml) was incubated for 3 hr at 25˚C with (DNA) or without (C)

*Figure 2. continued on next page*

*Figure 2. Continued*

2.5 µM DNA (DNA, 70b mbpTop1d oligonucleotide). Where indicated 0.5 mM thiram (Th), 1 mM dithiothreitol (DTT), 5 mM beta-mercaptoethanol (β-ME), of 10 mM glutathione (GSH) was added at the beginning of the incubation. The reaction was analyzed by gel electrophoresis and Coomassie staining and quantified by ImageJ.

The following figure supplements are available for figure 2:

**Figure supplement 1**. Modification of protein sulfhydryl groups by thiuram disulfides.

**Figure supplement 2**. Activation of Wss1 cleavage by APMA & Wss1 cysteines.

**Figure supplement 3**. Cleavage of Wss1-AQA mutant.

**Figure supplement 4**. Proposed mechanism for the regulation of Wss1 protease activity by cysteine switch mechanism.

---

DNA-induced Wss1 proteolysis. This result implies that, in cells, spontaneous activation of Wss1 by DNA may be inhibited by intracellular/nuclear thiols. Conversely, oxidative damage would be expected to deplete thiols and increase ssDNA lesions, resulting in activation of the latent protease activity of Wss1.

DNA may activate Wss1 in two ways (*Figure 2—figure supplement 4*). First, interaction of a positively charged WLM domain with DNA may induce conformational changes facilitating displacement from the active site of the negatively charged C-terminal peptide bearing the inhibitory cysteine. Additionally, DNA may facilitate Wss1 oligomerization and greatly promote proteolysis 'in-trans'. Both processes may be linked in a 'cooperative mechanism' (*Figure 2—figure supplement 4*). The initial activation events seem to be rate-limiting in the DNA-dependent pathway and may account for an observed lag phase in Wss1 cleavage (*Figure 2C*). Indeed thiram greatly accelerates DNA-dependent Wss1 proteolysis suggesting a cooperative interaction between '*in-cis*' and '*in-trans*' cleavage (*Figure 2B,C*).

## Wss1 processes poly-SUMO chains

Because it has been reported that Wss1 is a SUMO isopeptidase (*Mullen et al., 2010*) and we observed that it binds specifically polysumoylated substrates (*Figure 1D*), we examined whether it disassembles SUMO chains. Preformed chains were incubated with Wss1 in the presence of DNA or thiram, and the reaction products were analyzed by anti-SUMO western blot (*Figure 3*). While there was a reduction in the apparent molecular weight of the chains, we did not observe accumulation of low molecular weight SUMO species expected from chain disassembly and seen with Ulp1. Also, in contrast to Ulp1, Wss1 failed to cleave a SUMO1-FP isopeptide substrate (*Geurink et al., 2012*) and was unable to process a linear His6-Ub-SUMO-HA substrate (*Figure 3—figure supplement 1*), suggesting that Wss1 is not an isopeptidase. Instead, we found that Wss1 treatment increases the intensity of high molecular weight SUMO able to penetrate the separating gel (*Figure 3*). Because this reaction coincided with auto-proteolysis of Wss1 and was suppressed by OPA, it was probably catalyzed by Wss1 metalloprotease activity. Thus, it is likely that Wss1 proteolysis generated SUMO species that more easily entered the gel matrix and produced an apparent increase in SUMO signal.

Next, we analyzed the proteolysis reaction by SDS-PAGE with coomassie staining, providing a better estimate of the protein quantity than western blot (*Figure 3*, bottom panel). Wss1 was activated by thiram and its proteolytic activity was compared with SUMO isopeptidase Ulp1. Notably, Ulp1 cleavage clearly showed that the majority of HMW-SUMO species did not enter the gel and accumulated on the well bottom (*Figure 3*, lane 8). Wss1 treatment produced a distinct smear on the top of the gel suggesting more SUMO species entered the gel matrix. This corroborates the western blot data (*Figure 3*, top panel). The time course of HMW-SUMO processing was similar to Wss1 auto-proteolysis, both being inhibited by OPA. Interestingly, SUMO chains accelerated thiram-induced Wss1 auto-proteolysis (*Figure 3*, lane 2 and lane 6). Thus, similarly to DNA (*Figure 2B*) HMW-SUMO stimulates Wss1 cleavage, perhaps by promoting Wss1 oligomerization.

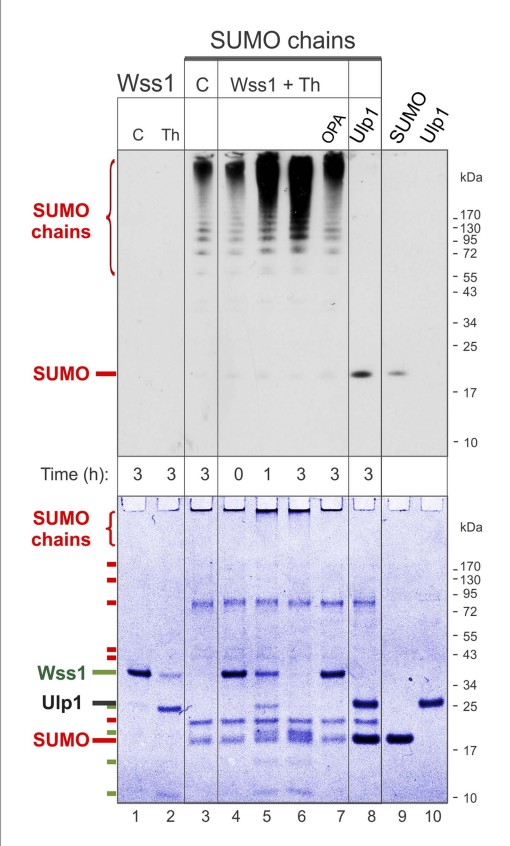

**Figure 3**. Processing of SUMO chains by Wss1. Comparative analysis of SUMO chains processing. Preformed SUMO-chains (200 µg/ml of SUMO monomer) were incubated with 100 µg/ml of recombinant HA-Wss1 (Wss1) or 100 µg/ml of Ulp1 for the indicated time at 25°C. Where indicated 0.5 mM thiram (Th) or 3 mM OPA was added. Control reactions (C) contain only Wss1 or polySUMO as indicated. The reaction was analyzed by western blotting with α-SUMO (top panel) and by gel electrophoresis with Coomassie staining (bottom panel). The positions of full-length HA-Wss1 (Wss1) and HA-Wss1 cleavage fragments are indicated in green. The positions of SUMO, SUMO chains, and SUMO E1–E3 components are shown in red.

The following figure supplement is available for figure 3:

**Figure supplement 1**. Wss1 cleavage of SUMO1-FP and His6-Ub-SUMO-HA substrates.

## Wss1 has a SUMO ligase-like activity

Wss1 has been reported to trim Ub from STUbL substrates (*Mullen et al., 2010*) so we asked if expression of Wss1 reduced the levels of ubiquitylated SUMO conjugates. Indeed, expression of Wss1 reduced the amount of ubiquitylated SUMO species in the cell but this reduction was independent of its metalloprotease active site (*Figure 4—figure supplement 1*). This suggests that the effect of Wss1 on levels of ubiquitylated SUMO conjugates could be due to competition with STUbL for binding of SUMOylated proteins.

Surprisingly, we found that Wss1 promoted sumoylation of cellular proteins (*Figure 4—figure supplement 1*). This effect was reproduced in vitro, where substoichiometric amounts of recombinant Wss1 catalyzed formation of SUMO chains (*Figure 4A*). In parallel, we observed that recombinant Wss1 and the WLM* mutant auto-sumoylate (*Figure 4B*). Because the reaction was SIM-dependent, it is likely that this E3-like ligase activity (*Figure 4A,B*) results from the juxtaposition of an E2~SUMO thiol ester and an acceptor SUMO due to polyvalent SUMO binding by Wss1 (*Merrill et al., 2010*).

The WLM* mutant that is unable to bind Zn repeatedly showed higher SUMO ligase activity compared to WT protein. To see whether the release of Zn increases sumoylation, we produced Wss1 apoprotein by OPA titration and analyzed its auto-sumoylation (*Figure 4C*). Increasing OPA concentration promotes sumoylation of WT-Wss1. This effect was not due to stimulation of SUMO E1–E2 enzymes because OPA did not increase the sumoylation of the WLM* mutant. At 3 mM OPA, WT, and WLM* formed a similar level of SUMO conjugates (*Figure 4C,D*). It seems therefore that extraction of Zn from metalloprotease active site somehow promotes Wss1-dependent sumoylation. Although the mechanism of this effect is not completely clear, it could be explained by our finding that OPA induces Wss1 oligomerization (*Figure 2B*). The oligomers can be seen by non-reducing gel electrophoresis and disappear in the presence of dithiothreitol (DTT) suggesting that they are

disulfide linked (*Figure 4D*). These data suggest that in the absence of Zn the regulatory cysteine 226 (*Figure 2—figure supplement 4*) induces intermolecular disulfide crosslinks, multiplication of SUMO-binding sites and increase in SUMO ligase activity (*Figure 4—figure supplement 2*).

In summary, our structural and biochemical analysis suggest that Wss1 is a SUMO-targeted minigluzincin-like metalloprotease that is regulated by a cysteine-switch mechanism. In its latent form, Wss1 binds and promotes SUMO chain elongation, while upon activation by DNA binding the enzyme self-cleaves and is capable of processing HMW-SUMO. Thus, Wss1 may function by synthesizing and/or binding and processing polysumoylated substrates.

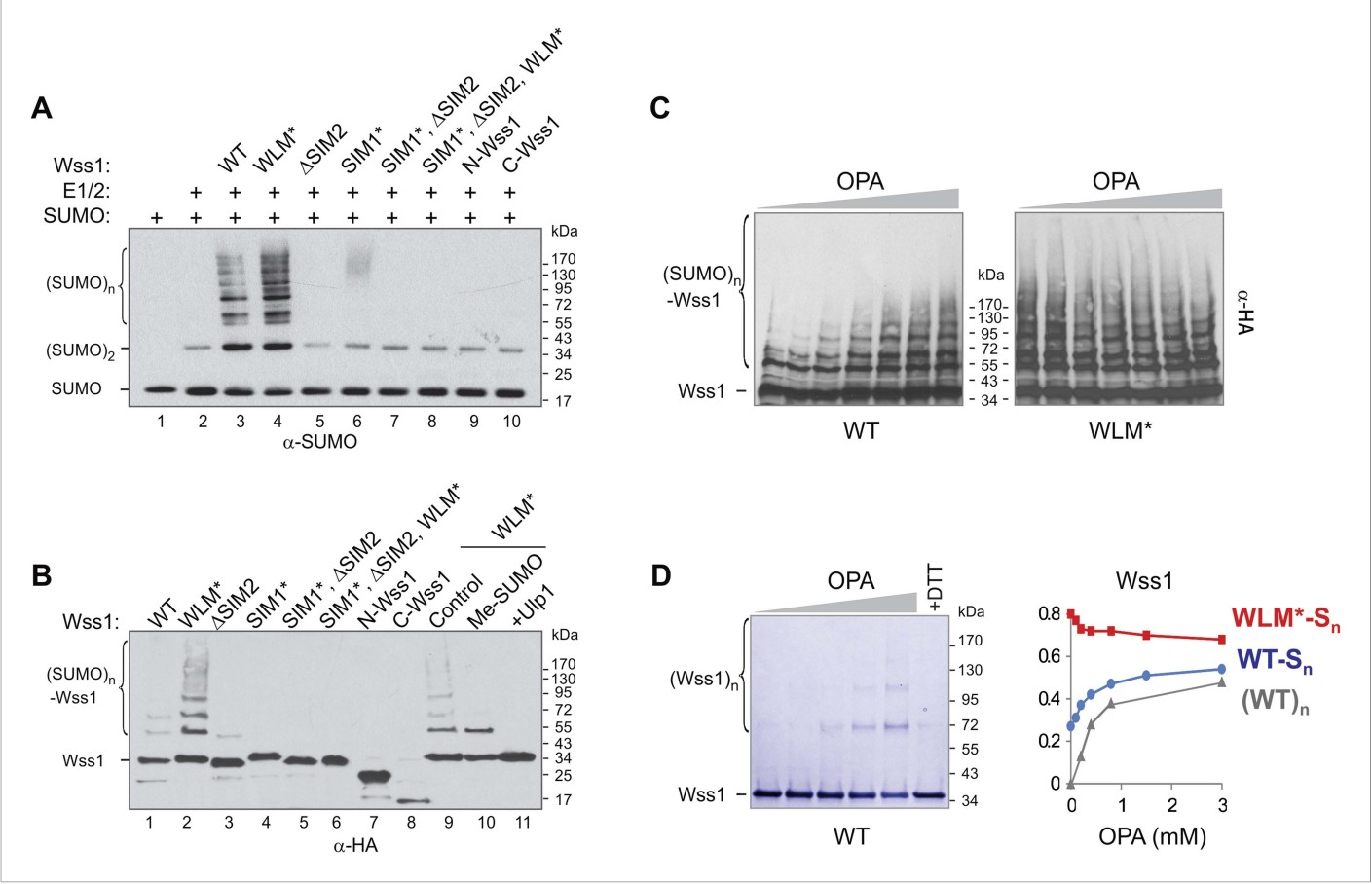

**Figure 4.** SUMO ligase activity of Wss1 protein. (**A**) Wss1 promotes SUMO chain formation. In vitro sumoylation reaction with HA-Wss1 constructs purified from bacteria ('Materials and methods'). The reaction was analyzed by western blotting with α-SUMO. (**B**) Auto-sumoylation of Wss1. In vitro sumoylation reaction with HA-Wss1 constructs analyzed by western blotting with α-HA. Lanes 9–11 show auto-sumoylation of WLM mutant. Lane 10: reaction with per-methylated SUMO. Lane11: Ulp1 was added after reaction. (**C**) Effect of OPA on Wss1 auto-sumoylation. The reactions were performed in the presence of increasing concentration of OPA (0–3 mM, see diagram below in *Figure 4D*). The level of Wss1-SUMO conjugates (Wss1-Sn) was analyzed by western blotting with α-HA and quantified by ImageJ. (**D**) OPA induces oligomerization of WT-Wss1. Recombinant HA-Wss1 protein (200 μg/ml) was incubated for 2 hr at 25°C with increasing concentration of OPA (0–3 mM, see diagram on the right). The reaction was analyzed by non-reducing gel electrophoresis and Coomassie staining and quantified by ImageJ. In the last lane, 200 mM DTT was added. Diagram on the right summarizes the effect of OPA on the level of sumoylated Wss1 (WT-Sn and WLM*-Sn) and Wss1 oligomers (WT)n. The data are normalized to the initial value at 0 mM OPA.

The following figure supplements are available for figure 4:

**Figure supplement 1**. Wss1 and STUbL may have common cellular substrates.

**Figure supplement 2**. Proposed mechanism for Wss1 SUMO-ligase activity.

## In vivo activities of WSS1

### Wss1 forms a SUMO-specific ternary complex with Cdc48 and Doa1

To further understand the regulation of ligase and protease activities of Wss1, we identified its cellular partners. Wss1 (chromosomally tagged with C-terminal Myc13) was immunoprecipitated from a cell lysate and bound partners (*Figure 5A*) were identified by mass spectrometry (MS). More than 50 proteins were connected in a network with STRING and classified using the gene ontology (GO) database (*Figure 5—figure supplement 1* and *Supplementary file 1*). Consistent with the involvement of Wss1 in DDR (*O'Neill, 2004*), the majority of binding partners belong to nucleic acid metabolic pathways involved in DNA repair and regulation of RNA polymerase. Surprisingly however, a third of the hits, many highly rated, were proteins implicated in intracellular transport and membrane associated processes.

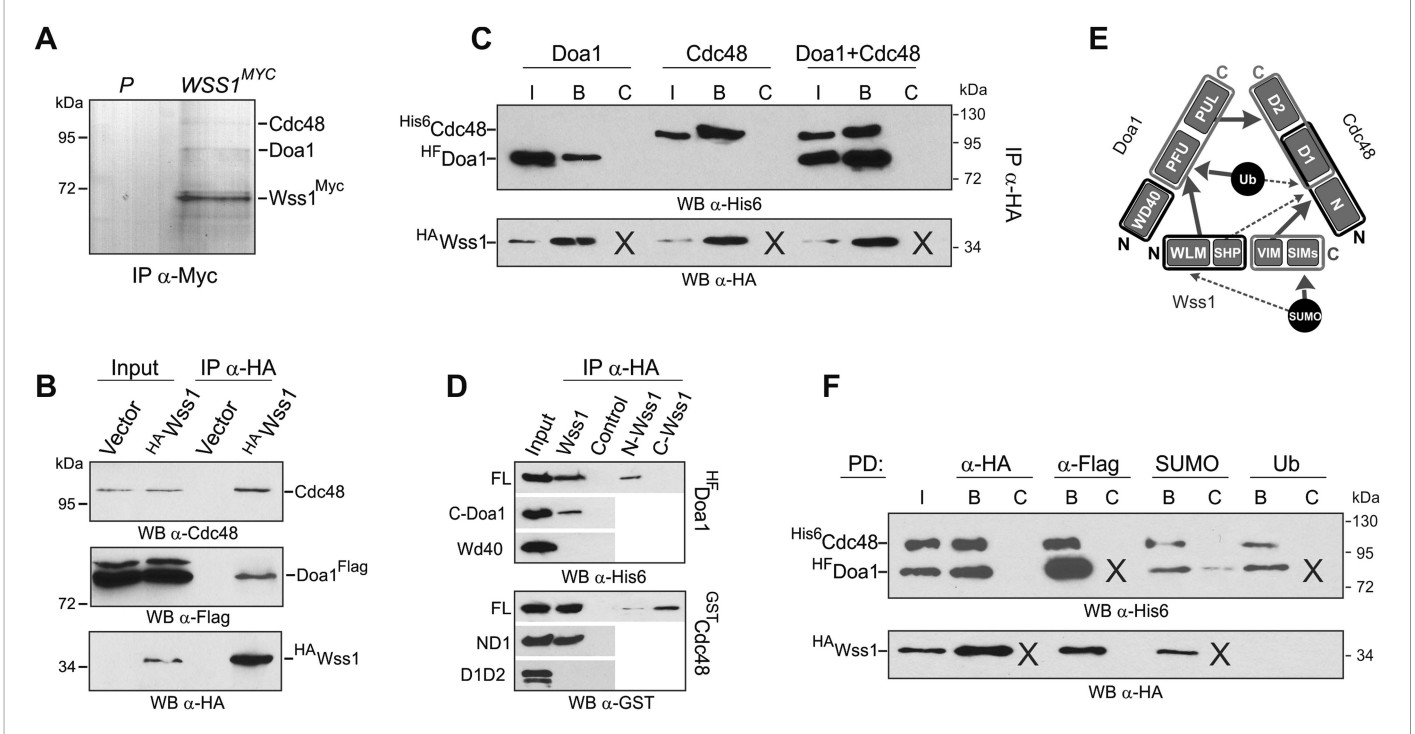

**Figure 5**. Wss1 forms SUMO-specific ternary complex with Cdc48 and Doa1. (**A**) Purification of Wss1-interacting proteins. Gel electrophoresis of the proteins isolated on anti-Myc beads from parental (P, BY4742) and *WSS1-MYC13* strains (MBY17, *Supplementary file 3*). The gel was stained with Coomassie blue. (**B**) Wss1 interacts with Cdc48 and Doa1. Cell lysates from *DOA1-FLAG* cells (MBY18, *Supplementary file 3*) transformed with HA-Wss1 or vector alone (*pYEPGAP-URA3*) were pulled down with anti-HA beads and analyzed by western blotting. Input: 10%. (**C**) Wss1 forms a ternary complex with Cdc48 and Doa1. HA-Wss1 protein was mixed with molar excess of His6-Cdc48 and his6-Flag (HF)-Doa1 and pulled down with anti-HA beads. The input (I) and bound (B) fractions were analyzed by western blotting. Control experiments (C) show protein binding in the absence of HA-Wss1 (crosses). Input: 20%. (**D**) Interaction within the Wss1/Cdc48/Doa1 ternary complex. The purified full-length (FL) or truncated versions of GST-Cdc48 and HF-Doa1 proteins were interacted with FL and truncated HA-Wss1 constructs. The proteins were pulled down with anti-HA beads and analyzed by western blotting. Input: 50%. (**E**) Schematic representation of protein interactions within Wss1/Cdc48/Doa1 complex. (**F**) Wss1/Cdc48/Doa1 ternary complex is SUMO specific. Stoichiometric amounts of HA-Wss1, His6-Cdc48, and HF-Doa1 proteins were mixed together and pulled-down with affinity beads. The ternary complex bound only to SUMO. In control experiments (C) either HA-Wss1 or HF-Doa1 protein was omitted (crosses). Input: 20%.

The following figure supplements are available for figure 5:

**Figure supplement 1**. Wss1 interactome.

**Figure supplement 2**. Characterization of Wss1-Cdc48-Doa1 interaction.

**Figure supplement 3**. Structural basis for Wss1-Cdc48 interaction.

The two most abundant Wss1 interactors were Doa1 and Cdc48. These interactions were further confirmed by immunoprecipitation of HA-Wss1 from *DOA1-FLAG* cell lysates (*Figure 5B*). Doa1 is a Cdc48 cofactor that binds the C-terminus of the chaperone through its PUL domain (*Mullally et al., 2006*). Because both Cdc48 binding motifs of Wss1, SHP, and VIM, should interact with the N-terminal domain of Cdc48 (*Yeung et al., 2008*; *Hanzelmann and Schindelin, 2011*; *Stapf et al., 2011*), we asked whether Wss1, Doa1, and Cdc48 form a ternary complex. Using purified recombinant proteins, we found that (i) Wss1 can bind Cdc48 and Doa1 separately (*Figure 5C* and *Figure 5—figure supplement 2A,B*), (ii) the proteins form a ternary 1:1:1 Wss1/Doa1/Cdc48 complex (*Figure 5C* and *Figure 5—figure supplement 2C*), and (iii) binding of Wss1 to Doa1 is specific, with few other Cdc48 cofactors observed in the pull-downs (*Figure 5—figure supplement 2A–C*).

To map the interactions within the Wss1/Doa1/Cdc48 complex, we performed a series of pull-down experiments with purified protein domains (*Figure 5D*). The results show that the N-terminus of

Wss1 binds the PFU domain of Doa1, while the SHP and VIM motifs of Wss1 interact with the N-terminus of Cdc48 (*Figure 5E*). Finally, we determined the NMR structure of the Wss1 VIM motif and used molecular docking and structure-based mutagenesis to identify R218, R219, F152 as key residues involved in Cdc48 binding (*Figure 5—figure supplement 3*). Mutation of all three of these residues (F2R) ablates Cdc48 binding (*Figure 5—figure supplement 3D*).

Previously, we demonstrated that Doa1 confers Ub specificity to Cdc48 complexes (*Mullally et al., 2006*). However, since Wss1 binds SUMO, and not Ub, we analyzed the specificity of the Wss1/Doa1/Cdc48 ternary complex. Pull-down experiments revealed that the ternary complex binds SUMO but not Ub, suggesting that the binding of Wss1 and Ub to Doa1 is mutually exclusive (*Figure 5F*). This result may be explained if the N-terminus of Wss1 actually adopts the predicted Ub-like beta-grasp fold and competes with Ub for binding to the Doa1 PFU domain.

Taking together, these data demonstrate that Wss1 is a Cdc48-interacting protein that partners with a Cdc48 cofactor, Doa1, to form a ternary Wss1/Doa1/Cdc48 complex. Although Doa1/Cdc48 has been considered to be Ub specific, we found that Wss1 redirects Doa1/Cdc48 to bind SUMO (*Figure 5F*, and *Figure 5—figure supplement 2E*). Moreover, we also observed limited SUMO binding by Doa1 alone (*Figure 5F*, and *Figure 5—figure supplement 2E*). In contrast to the Npl4/Ufd1/Cdc48 complex, which interacts with both Ub and SUMO (*Nie et al., 2012*), the Wss1/Doa1/Cdc48 ternary complex is SUMO specific.

## Cellular Wss1 sumoylates proteins and promotes their binding to Cdc48

We next examined the role of Wss1 in the regulation of SUMO metabolism by perturbing a variety of structural elements implicated in activity and binding. Wss1 proteins were expressed from a pYepGAP vector, and sumoylation of cellular proteins was analyzed by western blotting. Contrary to the accepted role of Wss1 as a protease, but consistent with its role as a SUMO ligase, Wss1 expression results in a marked buildup of high molecular weight SUMO conjugates at the top of the gel and two major species at 120 kD and 140 kD (*Figure 6A*). A similar effect was observed with WLM*, but not with the ΔSIM2 mutant, demonstrating that SUMO-binding and not protease activity was involved. None of the known SUMO ligases was required for stimulation of sumoylation by Wss1 (*Figure 6—figure supplement 1*) suggesting that the observed activity was due to Wss1 directly. Most strikingly, Siz1 appears to be responsible for most sumoylation under basal conditions, and Wss1 expression partially rescued the decreased SUMO conjugation seen in *siz1Δ* cells.

A major sumoylated cellular species observed upon Wss1 expression is a 120 kD SUMO-conjugate. To test if this band was derived from the Wss1 interactors, Doa1 (80 kD) and Cdc48 (92 kD), we expressed TAP-tagged Doa1 or Cdc48 and analyzed sumoylation profiles (*Figure 6B*). While the pattern of sumoylation in *DOA1-TAP* cells was similar to untagged strains, the major sumoylated band in *CDC48-TAP* cells was shifted upward by the expected 20 kD (the size of TAP-tag), suggesting that this conjugate was mono-sumoylated Cdc48. This was confirmed by anti-TAP western blot (*Figure 6B*). Sumoylation of Cdc48 by Wss1 required the SIM2 motif (*Figure 6B,C*). Wss1 mutants deficient in Cdc48 binding (R218/219S and F2R, *Figure 5—figure supplement 3D*) failed to sumoylate Cdc48, yet still produced HMW-SUMO conjugates. This finding suggests that sumoylation by Wss1 is E3-like and requires specific Wss1/substrate interactions. These cellular results were confirmed in vitro (*Figure 6—figure supplement 1*) where Cdc48, but not Doa1 were sumoylated by Wss1 and the proteolytic mutant WLM*. Consistent with our previous data (*Figure 4B*), WLM* was more highly sumoylated than the wild-type protein.

Since expression of Wss1 also induced the accumulation of other high molecular weight SUMOylated species, we asked whether these conjugates were bound to Cdc48. Various Wss1 constructs were expressed in a *CDC48-TAP* strain and proteins bound to immobilized Wss1 were isolated (*Figure 6D*). Wss1-beads retrieved Cdc48 and SUMO conjugates, including sumoylated Cdc48-TAP. Although similar amounts of HMW-SUMO conjugates were isolated from an F2R strain, it did not contain sumoylated Cdc48, confirming that Wss1 must bind Cdc48 to catalyze its sumoylation. Isolation of Cdc48-TAP complexes with IgG agarose also retrieved HMW-SUMO conjugates. These conjugates were specifically eluted from the IgG agarose by TAP-tag cleavage with TEV protease suggesting that these proteins were bound to Cdc48 (*Figure 6D*). The binding of HMW-SUMO to Cdc48 was Wss1-dependent since no SUMO-conjugates were co-purified with Cdc48 in the F2R-transformed strain.

Collectively, these data show that, in the absence of genotoxic stress, Wss1 sumoylates proteins with which it interacts. This E3-like activity depends on SIM2 and was reproduced in vitro (*Figure 4A,B*

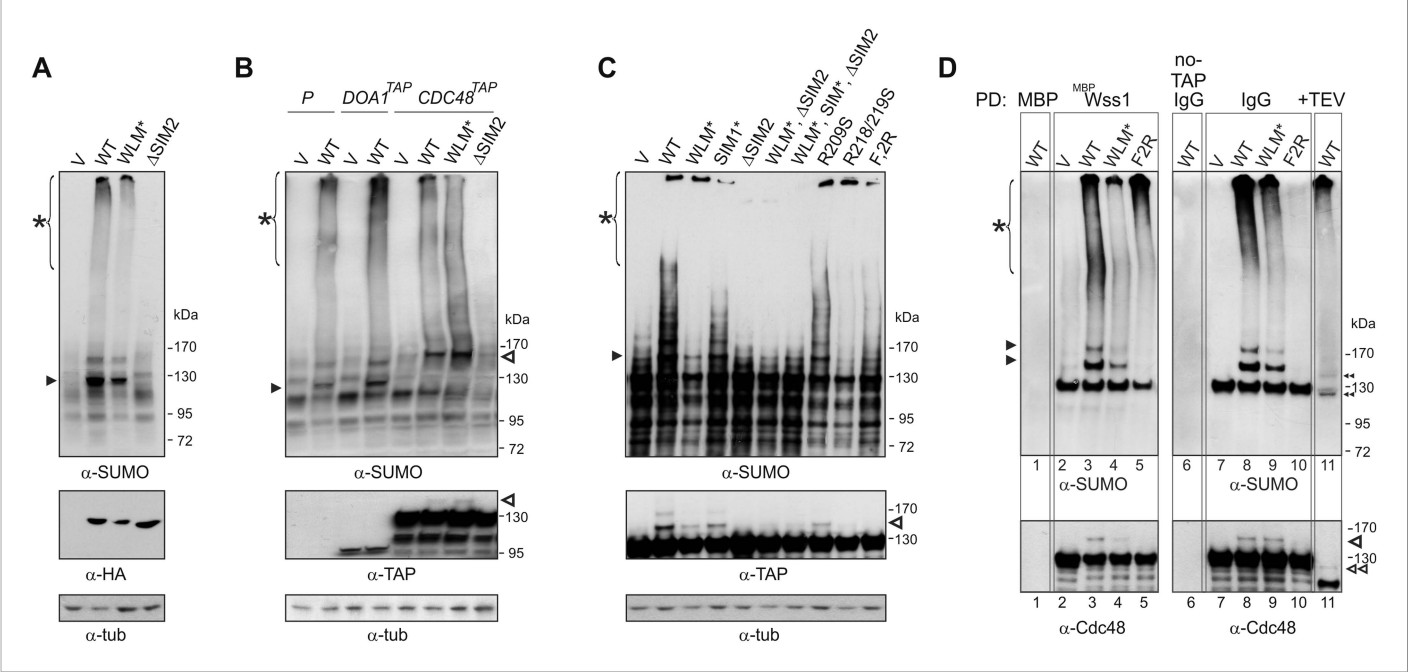

**Figure 6**. Wss1 sumoylates cellular proteins and promotes their binding to Cdc48. (**A**) Wss1 sumoylates cellular proteins. HA-Wss1 constructs were expressed from *pYEPGAP-URA3* vector (V) in BY4742 cells, and cell lysates were analyzed by western blotting. Abbreviations of the constructs are the same as in *Figure 1A*. The asterisk indicates HMW-SUMO, and the black arrowhead shows the 120 kD SUMO species. (**B**) Wss1 sumoylates Cdc48. Sumoylation was analyzed in parental (P, S288C) and *DOA1-TAP* and *CDC48-TAP* cells (*Supplementary file 3*) transformed with *pYEPGAP-URA3* constructs as in *Figure 6A*. The asterisk indicates HMW-SUMO, and the black arrowhead shows the 120 kD SUMO species. The white arrowhead shows the 150 kD SUMO-Cdc48-TAP species. (**C**) Effect of Wss1 mutations on Cdc48 sumoylation in CDC48-TAP cells. The cells were transformed with *pYEPGAP-URA3* constructs and analyzed by western blotting. The asterisk indicates HMW-SUMO, and the black arrowhead shows the 150 kD SUMO species. The white arrowhead shows the 150 kD SUMO-Cdc48-TAP species. (**D**) Wss1 promotes binding of SUMO conjugates to Cdc48. Lysates from *CDC48-TAP* cells transformed with Wss1 constructs were pulled down either with Wss1-beads (Wss1, 'Materials and methods') or IgG agarose and analyzed by western blotting. Control lanes show pull-down of *CDC48-TAP* cells lysate with MBP-beads (lane 1, MBP) and IgG immuno-precipitation of a lysate from non-tagged parental CDC48 cells (lane 6, no-TAP IgG, S288C). The last lane shows the eluate from WT-IgG beads (the same as lane 8) treated with 50 µg/ml TEV protease (lane 11, TEV). The asterisk indicates HMW-SUMO, the black arrowheads show the SUMO-Cdc48-TAP species, and 130 kD band is Cdc48-TAP cross-reacting with α-SUMO. The double black arrowheads show the SUMO-Cdc48 species after the cleavage of TAP-tag by TEV. The white arrowheads show SUMO-Cdc48 species before (simple) and after (double) TAP-tag cleavage by TEV.

The following figure supplement is available for figure 6:

**Figure supplement 1**. Wss1 promotes sumoylation of Cdc48 and other cellular proteins.

and *Figure 6—figure supplement 1B,C*). Thus, the finding that Wss1 promotes binding of HMW-SUMO by Cdc48 may be related to Wss1 function in the cell.

We next turned our attention to the role of Wss1 under two types of stress, SUMO stress induced by the *smt3-331* mutant allele and genotoxic stress induced by DNA damage.

## The Wss1-dependent decrease of HMW-SUMO and suppression of temperature-sensitivity in smt3-331 cells requires Wss1 proteolytic activity

It has been reported that Wss1 suppresses the temperature sensitivity (*ts*) of the SUMO-mutant allele *smt3-331* (*Biggins et al., 2001*). To ask if this suppression required the SUMO ligase or protease activity of Wss1, we probed the role of Wss1 in *smt3-331* cells. The *ts*-suppression required both correct SUMO binding via SIM2 and an intact protease domain, but not binding to Cdc48 mediated by R218, R219, and F152 of Wss1. The expression of a C-terminal Wss1-GFP fusion (at much lower levels) showed nearly the same suppression (*Figure 7A*). Looking at sumoylation, we found that *smt3-331* has negligible monomeric SUMO and an elevated level of HMW-SUMO conjugates, pointing to a defect in SUMO metabolism (*Figure 7A*). Wss1 induced a significant decrease in SUMO conjugates in *smt3-331* cells (*Figure 7A* and *Figure 9—figure supplement 1A*) but not in the wild-type *SMT3*

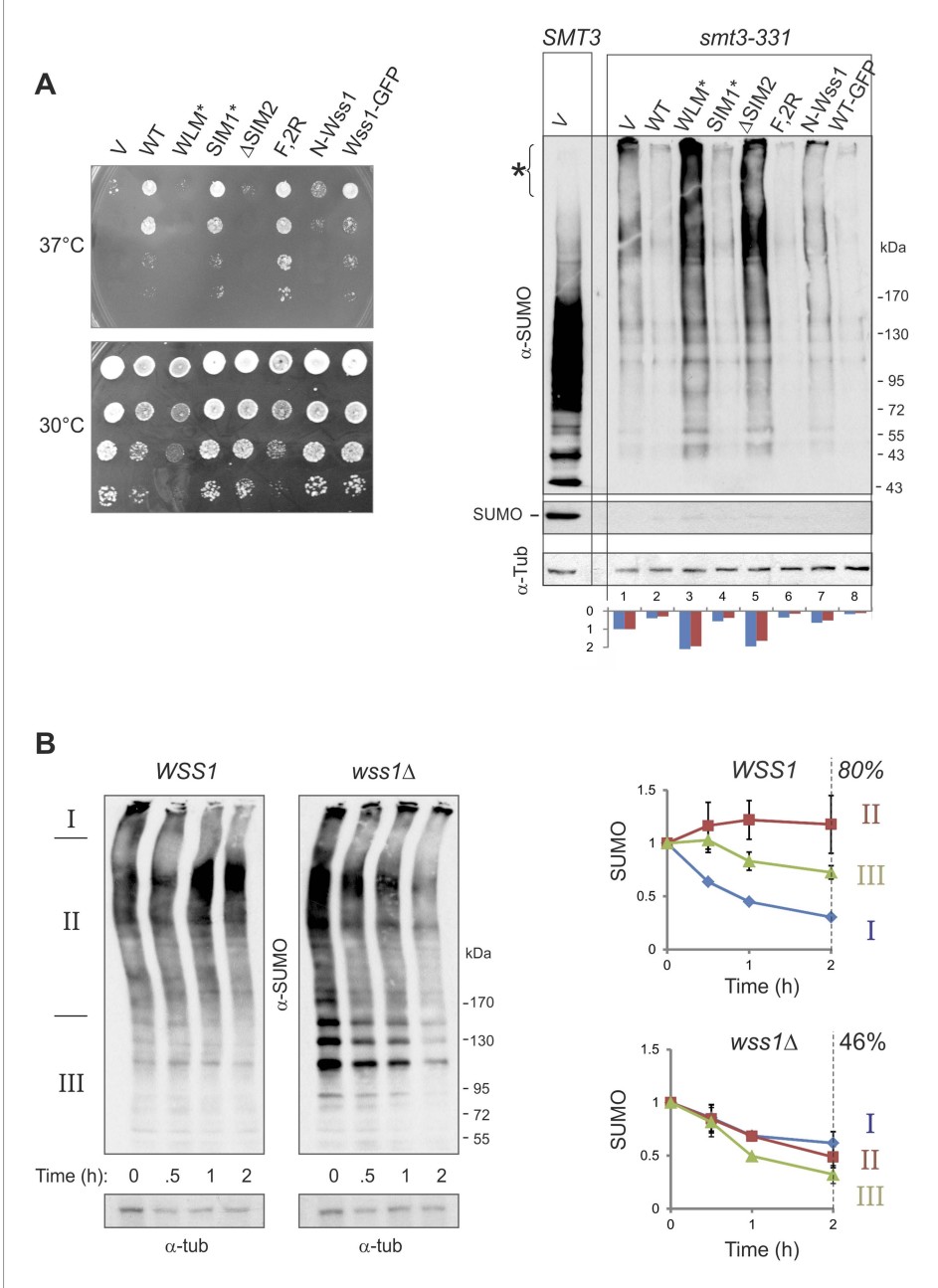

**Figure 7**. Wss1 regulates HMW-SUMO in *smt3-331* cells. (**A**) Suppression of *Smt3-331* phenotype by Wss1. Wss1 constructs were expressed in *smt3-331 wss1Δ* cells (MBY13, *Supplementary file 3*) as HA-fusions using *pYEPGAP-URA3* vector (V, etc) or as GFP fusion using *pUG35* vector (Wss1-GFP, *Supplementary file 4*). Left panel: ts-suppression assay on SD-ura plates. Right panel: western blot analysis of *SMT3* (first lane labeled *SMT3*, MBY14 strain) and *smt3-331* cell lysates with α-SUMO. Asterisk shows HMW-SUMO. Histogram: ImageJ quantification of total (blue) or HMW-SUMO (red) conjugates, normalized to the value obtained with vector alone (V). (**B**) Wss1 controls HMW-SUMO. Cellular protein synthesis was inhibited in exponentially growing *smt3-331 WSS1* (SBY331, *Supplementary file 3*) and *smt3-331 wss1Δ* cells (MBY13) by 50 µg/ml cycloheximide (CHX). WCL was analyzed by western blotting with α-SUMO. SUMO signal was measured by ImageJ in three zones I–III and normalized to the initial value. The plots show the kinetics of SUMO change (error bars represent standard deviations). The percentage on the right indicates total amount of SUMO signal remained after 2 hr.

The following source data is available for figure 7:

**Source data 1**. Western blot quantification for *Figures 7A* and *7B*.

strain (*Figure 6*). The effect of Wss1 mutants on sumoylation was similar to their effect on *ts*-suppression, suggesting a correlation between accumulation of HMW-SUMO and cell survival (*Figure 7A* and *Figure 9—figure supplement 1A*).

These data strongly suggest that Wss1 protease is directly involved in the metabolism (i.e., disassembly or degradation) of toxic HMW-SUMO in s*mt3-331* cells. To test this possibility, we compared the degradation kinetics of SUMO conjugates in *WSS1 smt3-331* and *wss1Δ smt3-331* strains (*Figure 7B*) we examined (I) very-HMW-SUMO at the top of the gel, (II) intermediate-HMW-SUMO of size >150 kD, and (III) SUMO conjugates below 150 kD. The *WSS1* strain had 1.5–2 times less total SUMO conjugates than *wss1Δ* (*Figure 7B* and *Figure 9—figure supplement 1A*). Upon addition of cycloheximide to inhibit protein synthesis, there was a progressive decrease in very-HMW-SUMO in the *WSS1* strain accompanied by an increase in intermediate-HMW-SUMO, while the SUMO conjugates of type III showed only slight decline. After 2 hr, only a small decrease in total SUMO conjugates had occurred (20%). In contrast, in the *wss1Δ* strain, very-HMW-SUMO was the most stable SUMO pool, while the other conjugates showed a gradual decrease resulting in a 54% decrease in total SUMO conjugates at 2 hr.

These data show that, when expressed from its endogenous loci, the protease activity of Wss1 processes specific very-HMW-SUMO and minimizes SUMO degradation. Increasing Wss1 expression using an exogenous promoter may shift cells toward degradation of all SUMO species. Interestingly, neither *ts*-suppression nor clearing of HMW-SUMO by Wss1 require binding to Cdc48 (*Figure 7A*).

Thus, Wss1 acts as a HMW-SUMO-processing protease in *smt3-331* but not in the *SMT3* strain. Several observations suggest that the proteolytic activity of Wss1 might be specifically activated in *smt3-331* cells. Thus, we found that in *smt3-331* strain the level of exogenously expressed HA-Wss1 and, particularly, HA-WLM* protein was greatly decreased in *WSS1* compared to *wss1Δ* cells pointing to a mechanism involving Wss1-mediated degradation (*Figure 8A*). Assuming similar rates of transcription, the massively higher level of ΔSIM2-Wss1 observed in the steady state suggests that this degradation was SUMO dependent. Therefore, consistent with our observations in vitro (*Figure 2* and *Figure 3*), Wss1 appears to undergo autoproteolytic activation and degradation stimulated by interaction with HMW-SUMO. Corroborating this conclusion, Wss1 protein was destabilized in *smt3-331* compared to *SMT3* cells and its degradation depended on the presence of the native protease and SIM domains (*Figure 8—figure supplement 1A*). At the same time, we found that *wss1Δ* increased the level of Cdc48 protein in *smt3-331* mutant, but not in *SMT3* strain (*Figure 8B*). In complementation assays, only WT Wss1 was able to reduce the level of Cdc48 protein in *wss1Δ* cells. These data suggest that Wss1 degrades itself and Cdc48 in *smt3-331* cells, and that this activity requires its protease domain, SUMO-interaction and Cdc48 binding. Because Wss1 binding to Cdc48 is not required for *ts*-suppression (*Figure 8A*), the degradation of Cdc48 by WT-Wss1 seems to be a sort of collateral damage induced by activation of the Wss1 protease by increased HMW-SUMO in *smt3-331* cells. Loss of Cdc48 itself would be detrimental to the cell and the lack of this degradation in F2R-Wss1 could explain why F2R-Wss1 is a better *ts*-suppressor than WT Wss1 (*Figure 8A*).

## Wss1 metalloprotease is activated upon genotoxic stress

*Smt3-331* mutants accumulate large-budded cells (*Biggins et al., 2001*) and have significantly increased frequency of gross chromosomal rearrangements reflecting ongoing DNA damage (*Lee et al., 2011*). We assume that the Wss1 protease is activated in *smt3-331* due to both the accumulation of ssDNA and SUMO conjugates induced by ongoing genotoxic stress.

To ask if direct DNA damage could activate Wss1, we treated cells with the UV mimetic 4-NQO and analyzed the steady-state amount of Wss1. In *WSS1* cells, 4-NQO induced significant reduction of exogeneously expressed Wss1 and WLM* proteins while leaving ΔSIM2 protein level unchanged (*Figure 8A*). The kinetics of Wss1 protein degradation confirmed that the observed effect was due to destabilization of Wss1 protein in 4-NQO-treated cells (*Figure 8—figure supplement 1A*). Cdc48 and WLM* are stable when *wss1Δ* cells are treated with 4-NQO, suggesting that DNA damage activates Wss1 and promotes its own and Cdc48 degradation in a SUMO-dependent mechanism.

To directly test if sumoylated proteins induced by 4-NQO are substrates for Wss1, HA-Wss1 protein complexes were isolated on anti-HA beads from w*ss1Δ* cells expressing HA-Wss1 and treated with 4-NQO (*Figure 8C*). All Wss1 constructs, except ΔSIM2, co-purified with SUMO-conjugates. Interestingly, all Wss1 complexes, except the F2R mutant (which cannot bind Cdc48) also contained ubiquitylated proteins (*Figure 8C*, middle panel). Because the presence of Ub-conjugates correlated

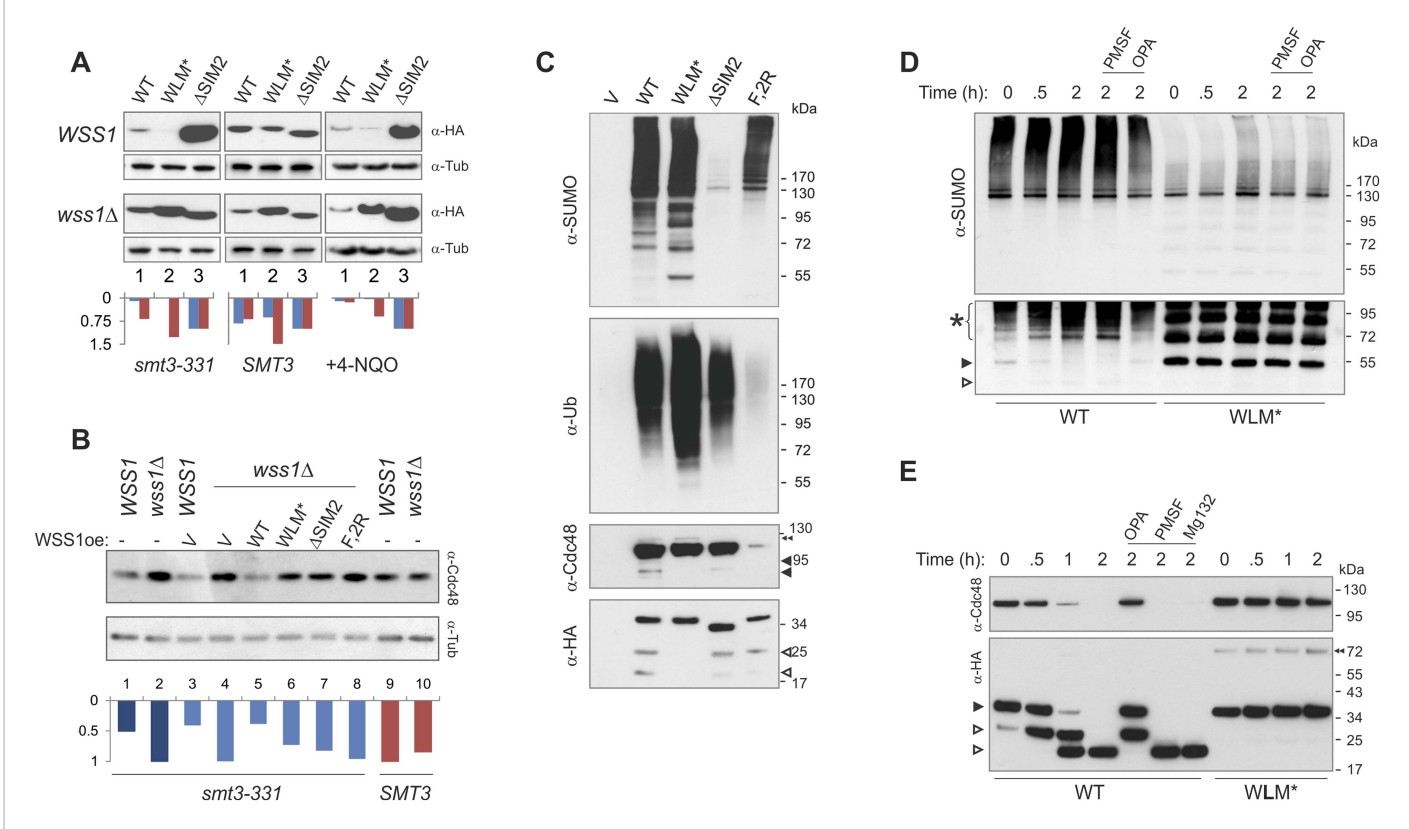

**Figure 8**. Wss1 protease is activated by DNA damage. (**A**) Wss1 degradation is stimulated by DNA damage. *WSS1* and *wss1Δ* cells (*smt3-331*, BY4742, and BY4742 + 0.2 µg/ml 4-NQO) were transformed with HA-Wss1 constructs (*pYEPGAP-URA3* vector) and grown at 30°C. The WCLs were analyzed by western blotting with α-HA and quantified by ImageJ. Histograms show the level of HA-Wss1 constructs in *WSS1* (blue) and *wss1Δ* (red) strains normalized to the value of ΔSIM2 protein. (**B**) Wss1 promotes Cdc48 degradation in *smt3-331* cells. *WSS1* and *wss1Δ* cells (*smt3-331* and BY4742 strains, *Supplementary file 3*) were transformed with HA-Wss1 constructs and grown at 30°C. The WCLs were analyzed by western blotting with α-Cdc48 and quantified by ImageJ. Histogram shows Cdc48 signal normalized to the maximum value. (**C**) Composition of the Wss1 complexes. The complexes were isolated with α-HA beads from WCL of *wss1Δ* cells (MBY15, *Supplementary file 3*) transformed with HA-Wss1 constructs and treated with 0.2 µg/ml 4-NQO. Pulled down proteins were analyzed by western blotting. The double black arrowhead shows SUMO-Cdc48, black arrowheads show Cdc48 fragments, and white arrowheads show Wss1 fragments. (**D**) Proteolysis of SUMO-conjugates by Wss1. HA-WT and HA-WLM* complexes were isolated with α-HA beads, incubated at 30°C with or without inhibitors, and analyzed by western blotting with α-SUMO. The lysis buffer contained 30 mM NEM. The bottom panel is taken at higher exposure. The asterisk indicates HMW-SUMO-derived fragments, the black arrowhead shows SUMO-Wss1, and the white arrowhead shows SUMO-Wss1 fragment. (**E**) Degradation of Cdc48 and self-proteolysis. The same conditions as in *Figure 8D*, except that α-Cdc48 and α-HA were used for western blotting. The black arrowhead indicates full-length Wss1, and white arrowheads show Wss1 fragments. The double black arrowhead shows WLM* dimers.

The following source data and figure supplement are available for figure 8:

**Source data 1**. Western blot quantification for *Figure 8—figure supplement 1A*.

**Figure supplement 1**. Wss1 proteolysis.

with Cdc48 binding, we assume that these species were bound to Cdc48. The WLM* complex showed the highest level of ubiquitinylated species. Previous study proposed that Wss1 cleaves mixed Ub-SUMO chains (*Mullen et al., 2010*), perhaps explaining the higher level of Ub conjugates in the protease-deficient WLM* complex. However, like others (*Stingele et al., 2014*) we could not detect such activity in Wss1 complexes using a specific Ub-SUMO substrate (*Figure 3—figure supplement 1B* and *Figure 8—figure supplement 1B*). On the other hand, as WT and ΔSIM2 complexes had the similar level of Ub-conjugates, it seems that the majority of these species were not sumoylated and interacted with Wss1 primarily through Cdc48. Cdc48 and Wss1 were partially fragmented in WT and ΔSIM2, but not WLM* cells (*Figure 8C*).

To directly measure Wss1-catalyzed proteolysis of these sumoylated proteins, we examined the stability of Cdc48, Wss1, and sumoylated proteins in purified WT and WLM* complexes. Similarly to what we observed for in vitro-processing of SUMO chains by recombinant Wss1 (*Figure 3*), the isolated WT protein complex accumulated sumoylated species of intermediate size as a result of the proteolysis of HMW-SUMO (*Figure 8D*). Proteolysis was likely not the result of cysteine, serine, or threonine proteases because of the presence of N-ethylmaleimide (NEM) in the lysis buffer and the lack of inhibition by PMSF. In contrast, OPA inhibited SUMO processing, suggesting that metalloprotease activity is involved. In the WLM* complex, an intense ladder of sumoylated protein was present and no degradation of sumoylated species was observed. Notably, both Cdc48 and Wss1 are processed in the WT complex (*Figure 8E*) in a reaction inhibited by OPA, but not by NEM or PMSF. Neither Cdc48 nor Wss1 proteolysis were affected by MG132 proteasome inhibitor, again pointing to metalloprotease activity. The degradation of Cdc48 coincided with the stepwise truncation of Wss1 protein (*Figure 8E*), reminiscent to what we observed with recombinant proteins (*Figure 2*). The size of the primary N-terminal Wss1 fragment (25 kD) corresponded to cleavage at V189 (*Figure 2*). The 25 kD N-terminal fragment underwent further truncation to a 20-kD protein. Sequence analysis suggests that this corresponds to cleavage near the GG sequence in the SHP box (*Figure 8—figure supplement 1C*). Because the WLM* mutant remained stable over time, it seems that this proteolysis was a result of Wss1 self-processing and reflected initial steps of protease activation (*Chakraborti et al., 2003*; *Gomis-Ruth, 2003*). As both Wss1 cleavage sites are conserved in WLM family members (*Iyer et al., 2004*), this may represent a general regulatory mechanism employed by WLM proteases. Wss1 self-cleavage produces an N-terminal activated protease and a C-terminal protein fragment which contains SIMs and VIM motifs and probably still bridges Cdc48 and SUMO-substrates. These mechanisms could serve for spatiotemporal control of the substrate processing, especially if the proteolytic fragments remain associated.

Altogether, these data demonstrate that DNA damage or a defective SUMO pathway triggers the activation of Wss1 metalloprotease resulting in Wss1 self-processing and degradation of associated proteins. Because Wss1 proteolysis in the cell is SUMO dependent, Wss1 acts as a SUMO-targeted metalloprotease.

## Wss1 protects cells from Top1-induced toxicity in the absence of Tdp1

What are the specific polysumoylated substrates processed by Wss1? A previous large-scale survey revealed significant negative genetic interaction between *WSS1* and tyrosyl-DNA phosphodiesterase (*TDP1*) (*Dixon et al., 2008*). Tdp1 cleaves stalled topoisomerase I (Top1) cleavage DNA complexes (Top1cc) formed upon DNA damage (*Pommier et al., 2006*). In the absence of Tdp1, the removal of Top1cc becomes SUMO dependent (*Heideker et al., 2011*; *Nie et al., 2012*). As this manuscript was being prepared, *Stingele et al. (2014)* reported that Top1cc was a direct substrate for the protease activity of Wss1. Except where noted, we agree with these authors and have confirmed their results. The *wss1Δ tdp1Δ* double mutant is very sick, sensitive to Top1 inhibitor camptothecin CPT (*Figure 9A*), partially cured by reducing Top1 levels (*Figure 9—figure supplement 1B*), and completely cured by expressing active WT Wss1 (*Figure 9E*). These findings indicate that, in the absence of Tdp1, Wss1 protease becomes critically important for Top1cc processing.

Extending these studies, we find accumulation of multiple SUMO conjugates (in addition to Top1cc [*Stingele et al., 2014*]) that correlates with synthetic sickness in *wss1Δ tdp1Δ* cells. Similar to UV-irradiated *wss1Δ* cells (*Figure 9—figure supplement 1C*), the double *wss1Δ tdp1Δ* mutants exhibit a significant increase in sumoylated proteins, both total and chromatin bound (*Figure 9B*). Because Wss1 protects cells from UV-induced DNA damage (*O'Neill, 2004* and *Figure 9—figure supplement 1C*) and Top1cc toxicity (*Stingele et al., 2014* and *Figure 9A,E*), one major function of this protein appears to be clearing a variety of toxic HMW-SUMO from the chromatin.

To demonstrate that Top1 is a Wss1 substrate in vivo, Wss1 was expressed in *TOP1-TAP wss1Δ* cells and Top1 was isolated on IgG beads. In a *wss1Δ* background, Top1 was sumoylated in a reaction requiring SUMO, but not Cdc48 binding (*Figure 9C*). Sumoylated Top1 and HMW-SUMO species were specifically eluted by TAP-tag cleavage with TEV protease suggesting that these conjugates were bound to Top1 (*Figure 9C*). In a *wss1Δ tdp1Δ* strain, expression of Wss1 reduced both Top1 levels as well as sumoylated Top1, suggesting protease activation (*Figure 9D*). Consistent with this, when WLM* was expressed, there was an even more severe growth defect and a strong accumulation of SUMO-Top1 as well as other SUMO species that might represent sumoylated Top1 fragments and/

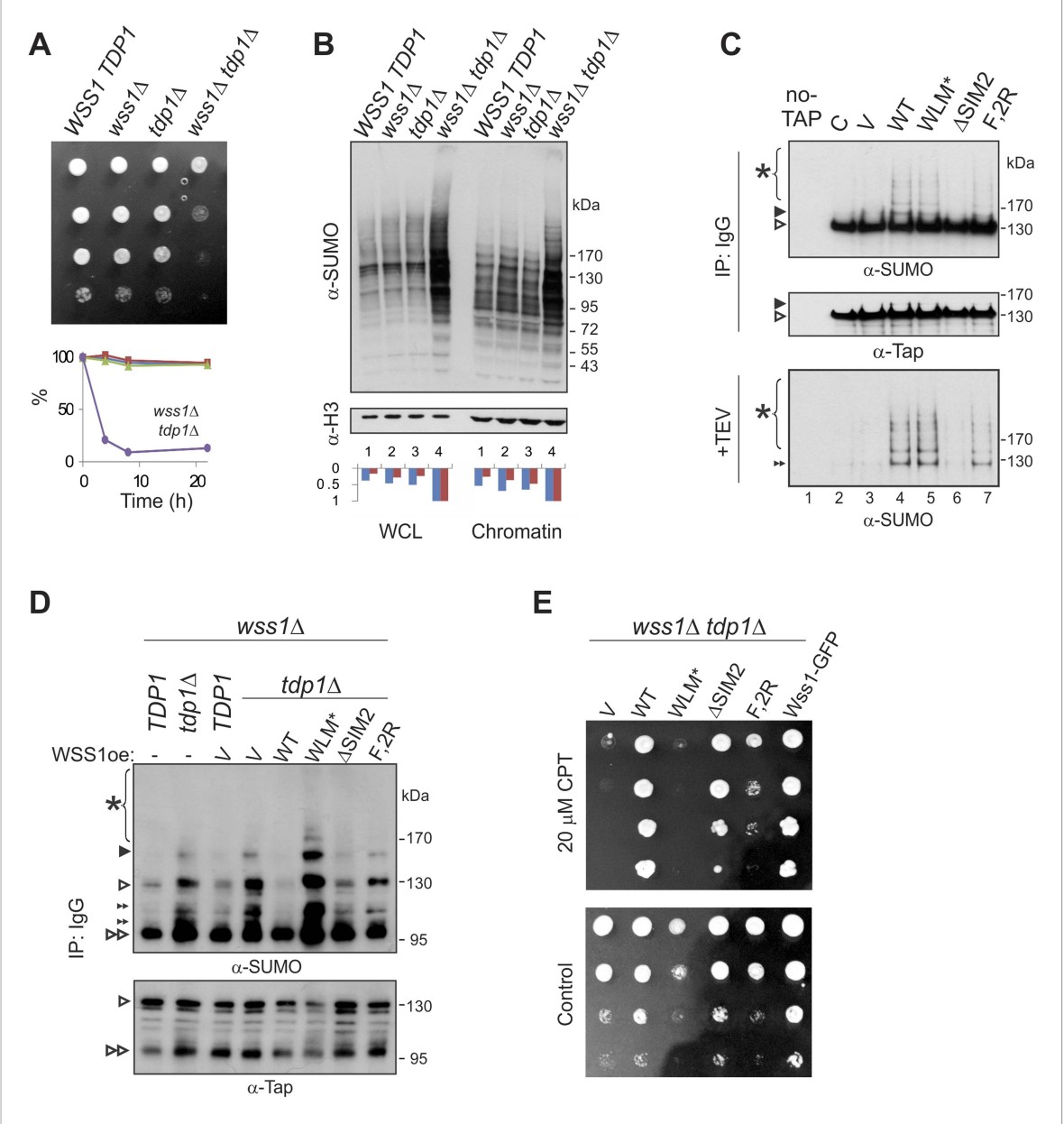

**Figure 9**. Wss1 processes Top1 topoisomerase. (**A**) Wss1 is required for normal growth of *tdp1Δ* cells. Same number of parental (BY4742) and mutant (MBY15, MBY31, and MBY37, *Supplementary file 3*) cells was spotted on YPD plates and incubated for 2 days at 30°C (without CPT). Diagram shows the viability of cells treated with 20 µM CPT for indicated times (estimated by colony-forming assay): *WSS1 TDP1* (red squares), *wss1Δ* (green triangles), *tdp1Δ* (blue diamonds), and *wss1Δ tdp1Δ* (magenta circles). (**B**) The *wss1Δ tdp1Δ* cells accumulate SUMO conjugates. WCL and chromatin fractions were analyzed by western blotting with α-SUMO. Asterisk shows HMW-SUMO. Histograms show ImageJ quantification of total (blue) or HMW (red) SUMO conjugates, normalized to the maximum value. (**C**) Wss1 promotes sumoylation of Top1. Top and central panels: Western blot analysis of Top1-TAP complexes isolated on IgG beads from *TOP1-TAP wss1Δ* cells (MBY35) transformed or not (C, control lane) with HA-Wss1 constructs (*pYEPGAP-URA3* vector). Control pull-down from non-tagged parental *TOP1* cells (S288C) is also shown (lane 1, no-TAP). Bottom panel: Western blot analysis of the eluates from IgG beads (the same as above) treated with 50 µg/ml TEV protease. Asterisk indicates HMW-SUMO. The black arrowheads show the position of SUMO-Top1-TAP, white arrowheads show Top1-TAP protein, and double black arrowhead shows SUMO-Top1 after cleavage by TEV. (**D**) Accumulation of sumoylated Top1 in *wss1Δ tdp1Δ* cells. Top1-TAP was isolated on IgG beads from *TOP1-TAP wss1Δ* (MBY35) or *TOP1-TAP wss1Δ tdp1Δ* (MBY39) cells transformed or not with HA-Wss1 constructs (*pYEPGAP-URA3*, V indicates vector alone) and analyzed by western blotting. Asterisk shows HMW-SUMO, white arrowheads show the position of Top1-TAP overlapping with one of the SUMO species, black arrowhead: SUMO-Top1-TAP, double black arrowheads: SUMO species and double white arrowheads: Top1-TAP fragment. (**E**) Suppression of *wss1Δ tdp1Δ* sickness by Wss1. The *wss1Δ tdp1Δ* cells (MBY39) were transformed with Wss1 constructs, spotted on SD- ura plates with or without 20 µM CPT and incubated at 30°C.

*Figure 9. continued on next page*

*Figure 9. Continued*

The following figure supplement is available for figure 9:

**Figure supplement 1**. Wss1 protects cells from genotoxic threats.

or associated proteins. Wss1 suppressed CPT-sensitivity of *wss1Δ tdp1Δ* cells (*Figure 9E*). Unexpectedly, ΔSIM2 was also a suppressor, while F2R had much weaker effect. Thus, Wss1 overexpression compensates for low SUMO and, albeit to a much lesser extent, Cdc48 binding (*Figure 9E*).

These data suggest that Wss1 functionally interacts with Top1, and this interaction is essential for clearing Top1-associated SUMO conjugates produced as a result of Tdp1 deficiency. Thus, in vivo evidence reported here confirms that Wss1 is directly involved in cleaving Top1cc complexes in a SUMO-dependent fashion.

## Wss1 localizes to the vacuole upon DNA damage

To examine the site(s) of Wss1 action, we analyzed cellular localization of this protein. GFP-Wss1 showed nuclear staining with an occasional single spot per cell nucleus as previously reported (*Figure 10A* and *Figure 10—figure supplement 1A*) (*van Heusden and Steensma, 2008*). Notably, when examined in *tdp1Δ* cells, GFP-Wss1 showed multiple foci, reflecting, probably, multiple sites of DNA damage (*Figure 10A*). Consistent with this conclusion, treatment of *TDP1* cells with the UV mimetic 4-NQO also induced multiple Wss1 foci (*Figure 10—figure supplement 1A*). The foci were not observed with the GFP-Wss1-ΔSIM2 construct, suggesting SUMO-dependent targeting of Wss1 to these foci (*Figure 10A*).

To test the role of Top1 in Wss1 relocalization to multiple foci seen in *tdp1Δ* cells, we constructed a strain in which the endogenous Top1 promoter was replaced by GAL promoter, and Wss1 was chromosomally tagged with GFP. Unexpectedly, in the absence of Tdp1, stimulating Top1 expression with GAL promoter resulted in accumulation of Wss1-GFP within the vacuole (*Figure 10B* and *Figure 10—figure supplement 1C*). Concomitant FACS analysis revealed ~twofold increase in Wss1-GFP fluorescence in *tdp1Δ* cells upon Top1 induction. Furthermore, many cells contained ~200 nm fluorescent punctate structures within the cytoplasm and on the vacuolar membrane suggesting the activation of the autophagic pathway (*Figure 10B* and *Figure 10—figure supplement 1C*). Notably, we also observed Wss1 translocation to the vacuole in 4-NQO- treated cells and *smt3-331* mutant (*Figure 10—figure supplement 1B*). Though preliminary, these results support the role of vacuole in Wss1 function in SUMO-dependent DDR.

## Discussion

Sumoylation of nuclear proteins, replication enzymes and DDR components, orchestrates assembly and disassembly of chromatin complexes. Yet, how SUMO-conjugates are processed remains largely unknown. Here we add critical detail to the metalloprotease Wss1-mediated DNA repair pathway involving SUMO-targeting and Cdc48-mediated depletion of sumoylated proteins from chromatin.

### Wss1 is a SUMO-ligase and a SUMO-targeted metalloprotease activated by genotoxic stress

Wss1 is a DDR component important for survival upon UV irradiation (*O'Neill, 2004*). Wss1, along with other proteins of WLM-family (Wss1-Like Metalloproteases), were predicted to be proteases (*Iyer et al., 2004*) and while this article was in preparation, *Stingele et al. (2014)* reported that Wss1 is a DNA-activated metalloprotease that processes DNA-protein crosslinks. Neither they, nor we, can confirm the SUMO- or Ub-isopeptidase activity previously observed (*Mullen et al., 2010*), and it seems plausible that it resulted from co-purified deubiquitinating enzymes. Our structural analysis suggests that WLM domains share similar structural organization with minigluzincin proteases (*Lopez-Pelegrin et al., 2013*) and SprT enzymes (*Figure 1—figure supplement 2*, see below).

Recombinant Wss1 produced in bacteria is largely insoluble. When purified under native conditions, soluble protein shows SUMO ligase activity (see below) while protease activity (*Figure 2*) required denaturation and refolding of inclusion bodies. *Stingele et al. (2014)* reported

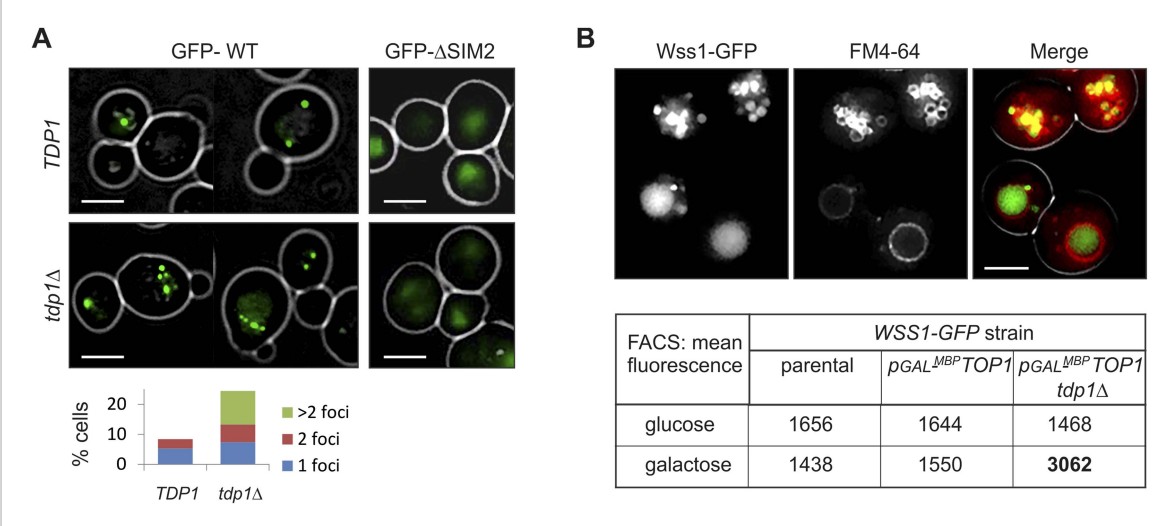

**Figure 10**. Wss1 localization upon DNA damage. (**A**) Increased incidence of GFP-Wss1 foci in *tdp1Δ* cells. Live cells (BY4742 and MBY31 strains) expressing GFP-Wss1 WT and ΔSIM2 constructs (*pUG36* vector with inducible *MET25* promoter, **Supplementary file 4**) were analyzed by fluorescence microscopy in SD-Ura-Met medium. Histogram shows percentage of cells with Wss1 foci in TDP1 (n = 220) and tdp1Δ (n = 342) cells. Scale bar is 2 μm. (**B**) Top1-induced stress in *tdp1Δ* cells induces accumulation of Wss1-GFP in vacuole. Localization of Wss1 protein was studied in cells expressing Wss1-GFP from endogenous loci (MBY24, MBY46, and MBY47 strains, **Supplementary file 3**). In *pGAL-MBP-TOP1* cells *TOP1* promoter was replaced by *GAL* promoter and Top1 was expressed as MBP-Top1 fusion. Top panels show *pGAL-MBP-TOP1 WSS1-GFP tdp1Δ* cells (MBY47) upon induction of MBP-Top1 expression in galactose medium. Vacuole was stained with FM4-64 dye. Scale bar is 2 μm. The table shows mean GFP fluorescence of parental (MBY24) and *pGAL-MBP-TOP1* cells (MBY46, and MBY47) measured by FACS (n = 20,000). The cells were grown in glucose or galactose medium at 30°C.

The following source data and figure supplement are available for figure 10:

**Source data 1**. IF microscopy and FACS data for Wss1 expression in TDP1 and tdp1Δ cells.
**Source data 2**. IF microscopy data for Wss1 localization upon DNA damage.
**Figure supplement 1**. Wss1 localization upon DNA damage.

that the protease activity of Wss1 is activated in vitro by any DNA longer than 16 bp. We too see activation by DNA, albeit only when ssDNA is present. We assume that differences in the refolding conditions lead to these subtle differences in the activity of recombinant protein. However, it is important to note that Wss1 protease activity is normally latent, but activated by genotoxic stress in cells.

Our principal insight into the Wss1 mechanism comes from the finding that the protease is regulated by a cysteine-switch mechanism. We demonstrated that Wss1 proteolysis can be activated by thiol-modifying reagents such as thiram and APMA as well as by ssDNA (**Figure 2** and **Figure 2—figure supplement 2**). In addition, polySUMO binding enhances the proteolysis of recombinant protein (**Figure 3**) and SIM2 is required for both Wss1 and Cdc48 cleavage under conditions of stress. Thus, binding to DNA is not absolutely required for Wss1 proteolytic activity. Moreover, our observations that Wss1 can process itself, sumoylated proteins and Cdc48 demonstrate that Wss1 substrates are not limited to DNA-bound proteins, as previously suggested by Stingele et al. (2014). DNA (**Figure 2**) and SUMO chains (**Figure 3**) may play a cofactor role by inducing Wss1 oligomerization and bringing together the protease and its substrates (**Figure 2—figure supplement 4**). Our data in vitro and in vivo strongly suggest that polysumoylated proteins are the substrates for Wss1. Although the exact mechanism of how Wss1 processes SUMO conjugates remains unclear, one possible scenario may be the activation of metalloprotease and cleavage within a hinge region of the polysumoylated proteins. Alternatively, our finding that Wss1 binds to SUMO-PA, self-cleaves at C-terminus and produces a ladder of proteolytic fragments upon activation by ssDNA may suggest that it functions as a SUMO-targeted carboxypeptidase.

Even more surprising was our finding that Wss1 promotes SUMO chain formation in vitro and in unstressed cells, acting as a SUMO ligase. Wss1 shows a preference for binding polysumoylated ligands (*Figure 1D*), sumoylates itself and associated cellular proteins, changes the global sumoylation pattern, and partially restores SUMO-conjugates in a *siz1Δ* strain. Sumoylation depended on tandem SIMs at the C-terminus of Wss1 and is probably the result of juxtaposed SUMO binding sites that bring SUMO thiol ester in proximity to an acceptor SUMO. Interestingly, among other known non-canonical SUMO E3-ligases, including RanBP2, polycomb Pc2, topors, Slx5, and Rad18, all, except RanBP2, catalyze sumoylation by SIM-dependent mechanism (*Ii et al., 2007*; *Perry et al., 2008*; *Merrill et al., 2010*; *Parker and Ulrich, 2014*). Furthermore, the oligomerization of Wss1 may also play an important role in SUMO ligase activity (*Figure 4D* and *Figure 4—figure supplement 2*). The extension of a SUMO-chain by Wss1 may signal assembly of Wss1 complexes required for substrate processing, as well as target these complexes to specific cellular locations. Notably, the sumoylation of Cdc48 was shown to induce its translocation from nucleus to membranous cytoplasmic structures (*Makhnevych et al., 2009*).

Finally, the role of SUMO binding is controversial. *Stingele et al. (2014)* suggested that SUMO binding is dispensable for Wss1 function although they did not demonstrate Wss1 proteolysis in the cell. We show that Wss1-dependent proteolysis is activated under conditions that damage DNA and is SUMO dependent. Although we too observed that Wss1-ΔSIM2 could suppress CPT sensitivity of *wss1Δ tdp1Δ* cells, competent SUMO binding is required for the clearing of HMW-SUMO conjugates as well as Wss1 targeting to *tdp1Δ*-induced foci. Moreover, because cellular Wss1-ΔSIM2 remained stable under various DNA-damaging conditions, it seems that SUMO binding is essential for Wss1 auto-proteolysis as well. The partial effects seen with SIM mutants may be masked by the fact that, as we have shown, the WLM domain also contributes to SUMO binding.

## Wss1, Cdc48, Doa1, and SUMO-dependent DDR

We find that full-length Wss1 interacts with Cdc48 ATPase (*Figure 5*, *Figure 5—figure supplement 2*, *Figure 5—figure supplement 3*, and *Supplementary file 1*), supporting a role for Wss1 in removing sumoylated proteins from the chromatin. Wss1 contains VIM and SHP Cdc48-binding motifs, conserved within the WLM family. Moreover, the incidence of other Cdc48-relevant interaction domains in WLM members indicates that these proteins might also be Cdc48 partners (*Iyer et al., 2004*). Our structural analysis demonstrated that both VIM and SHP motifs contributed to Cdc48 binding, suggesting a bipartite interaction mechanism, a characteristic mode of binding of common Cdc48 adaptors (*Bruderer et al., 2004*; *Yeung et al., 2008*). We also found that Wss1 specifically binds another Cdc48 cofactor, Doa1, and forms a ternary 1:1:1 Cdc48/Wss1/Doa1 complex. Although Doa1 and Cdc48 have been considered to be Ub specific (*Mullally et al., 2006*), we demonstrated here that Wss1 redirects the Doa1/Cdc48 complex to SUMO substrates.

Another Cdc48 complex, Cdc48/Ufd1/Npl4, binds mixed Ub-SUMO substrates and assists STUbL in SUMO-dependent Top1cc repair (*Nie et al., 2012*). Existing data suggest that Wss1 and STUbL may share common substrates and act in the same DDR pathway (*Figure 4—figure supplement 1* and *Mullen et al., 2011*). Indeed, their response to various genotoxic threats places Wss1 and STUbL in the same 'nucleotide-excision-repair' (NER) epistasis group. We identified NER proteins by MS suggesting physical association of Wss1 and NER machinery (*Figure 5—figure supplement 1* and *Supplementary file 1*). Both Wss1 and STUbL are involved in Top1cc repair in the absence of Tdp1, pathways shown to be NER-dependent (*Figure 9* and *Liu et al., 2002*; *Vance and Wilson, 2002*; *Heideker et al., 2011*; *Stingele et al., 2014*). Although the function of Wss1 and STUbL in NER is unclear, it may be required to reduce the size of NER substrates (*Stingele et al., 2014*). NER cannot process DNA adducts containing proteins larger than 16 kDa (*Nakano et al., 2007, 2009*), while the size of Top1 protein exceeds 90 kDa. Wss1 and STUbL may also act by removing stalled sumoylated NER enzymes in addition to NER substrates. Recent data demonstrate that DNA damage induces extensive sumoylation of multiple NER components (*Cremona et al., 2012*; *Psakhye and Jentsch, 2012*). Although it was suggested that SUMO assists the assembly of repair complexes, in some cases sumoylation promotes protein dissociation and turnover (*Hardeland et al., 2002*; *Fernandez-Miranda et al., 2010*; *Sarangi et al., 2014*).

Wss1 and STUbL pathways may direct substrates to different downstream degradation events, orchestrated by different Cdc48 complexes. The major degradative machineries of the cell are the

proteasome and the vacuole/lysosome, Cdc48 being implicated in both (*Meyer et al., 2012*). Chromatin-associated proteolysis involving STUbL pathways has been linked mostly to the Cdc48/ Ufd1/Npl4 complex and the proteasome (*Meyer et al., 2012*; *Vaz et al., 2013*; *Sriramachandran and Dohmen, 2014*). Alternatively, autophagy is implicated in Wss1-mediated processes; a third of our best-rated MS hits were proteins associated with vesicle transport and the vacuole, indicating a possible role of Cdc48/Wss1/Doa1 complex in vacuolar degradation. Doa1 has been implicated in regulation of two main vacuolar pathways, autophagy and vesicle sorting (*Ren et al., 2008*; *Ossareh-Nazari et al., 2010*). We have demonstrated that various genotoxic conditions induce accumulation of Wss1 within the vacuole. Moreover, under genotoxic stress Wss1 was found associated with autophagic vesicle-like structures in the cytoplasm and on the vacuolar membrane. Because autophagy is one of the DDR activated pathways (*Robert et al., 2011*; *Dotiwala et al., 2013*; *Vessoni et al., 2013*), our data suggest an intriguing possibility that Cdc48/Wss1/Doa1 mediates degradation of sumoylated chromatin substrates via specific autophagic mechanisms.

## A model for the role and regulation of Wss1 in DDR

Our data suggest a model for the SUMO-dependent extraction of proteins from chromatin (*Figure 11*). DNA lesions induce exposure of ssDNA and sumoylation of nearby chromatin-bound proteins (*Psakhye and Jentsch, 2012*). Thus, exposure of ssDNA and/or initial sumoylation of proteins at the sites of DNA damage could provide binding sites for Wss1. Its intrinsic ligase activity could then sumoylate other proteins, extending SUMO chains, become proteolytically activated. Wss1 would also recruit Doa1/Cdc48 to disassemble proteins from the damage site(s) and target the complexes to the vacuole. The proteolytic activation of Wss1 both down regulates Wss1 mediated action and fragments other proteins to induce their dissociation. Thus, we propose that autophagy processes poor proteasome substrates formed as a result of DNA damage, that is, protein–protein and protein–nucleic acid crosslinks, irreversibly trapped enzymes and abnormal nucleic acid intermediates (*Figure 11*). Alternatively, or in addition, STUBL could be recruited to ubiquitinate chromosomal lesions that have not been repaired and recruit Doa1/Ufd1/Npl4 to chaperone ubiquitinated substrates to the proteasome.

## Higher eukaryotes: DVC1/Spartan

Wss1 shows structural and functional similarity to DVC1/Spartan (*Figure 1—figure supplement 1C* and *Figure 1—figure supplement 2*), a protein from higher eukaryotes that brings Cdc48 to stalled replication forks (*Centore et al., 2012*; *Davis et al., 2012*; *Mosbech et al., 2012*). Like Wss1, DVC1/ Spartan protects cells from UV damage, probably by removing promiscuous TLS-polymerases from the chromatin. DVC1/Spartan has a putative metalloprotease domain SprT appended to a SHP motif, a PCNA-interacting PIP box and a Ub-binding Zn-finger domain UBZ4. No proteolytic activity has been observed for the proteins of the SprT family, yet the putative protease active site is essential for DVC1/Spartan function (*Kim et al., 2013*). Mutations in DVC1/Spartan SprT domain result in genomic instability (*Lessel et al., 2014*; *Maskey et al., 2014*) and have recently been linked to a new progeroid syndrome with early onset of hepatocellular carcinoma (*Lessel et al., 2014*). Our analysis suggests that the structure of SprT is very similar to the WLM domain (*Figure 1—figure supplement 1C* and *Figure 1—figure supplement 2*). Moreover, DVC1/Spartan has several cysteine residues, one of which, conserved C205, seems not to be involved in disulfide bonds and is a potential candidate for protease regulation by a cysteine-switch mechanism. Thus, DVC1/Spartan may also be a functional protease that assists Cdc48 in removing chromatin components upon DNA damage stress. While Wss1 is targeted to SUMO, DVC1/Spartan binds PCNA and Ub, reflecting probably the differences in DDR regulation in higher and lower eukaryotes. Notably, in DVC1/Spartan from *Caenorhabditis elegans* and other species PIP box is replaced by SIM upstream of UBZ4 domain, while some WLM proteins have both SIM and NZF (*Figure 1—figure supplement 1C*). This suggests that these proteins could have dual Ub-SUMO binding specificity and that in higher eukaryotes the SIM function is substituted by PIP motif.

## Conclusions

Overall, this work confirms and extends the elegant findings of *Stingele et al. (2014)*. It reveals the regulation of Wss1 activity by a cysteine-switch mechanism, demonstrates a ternary complex

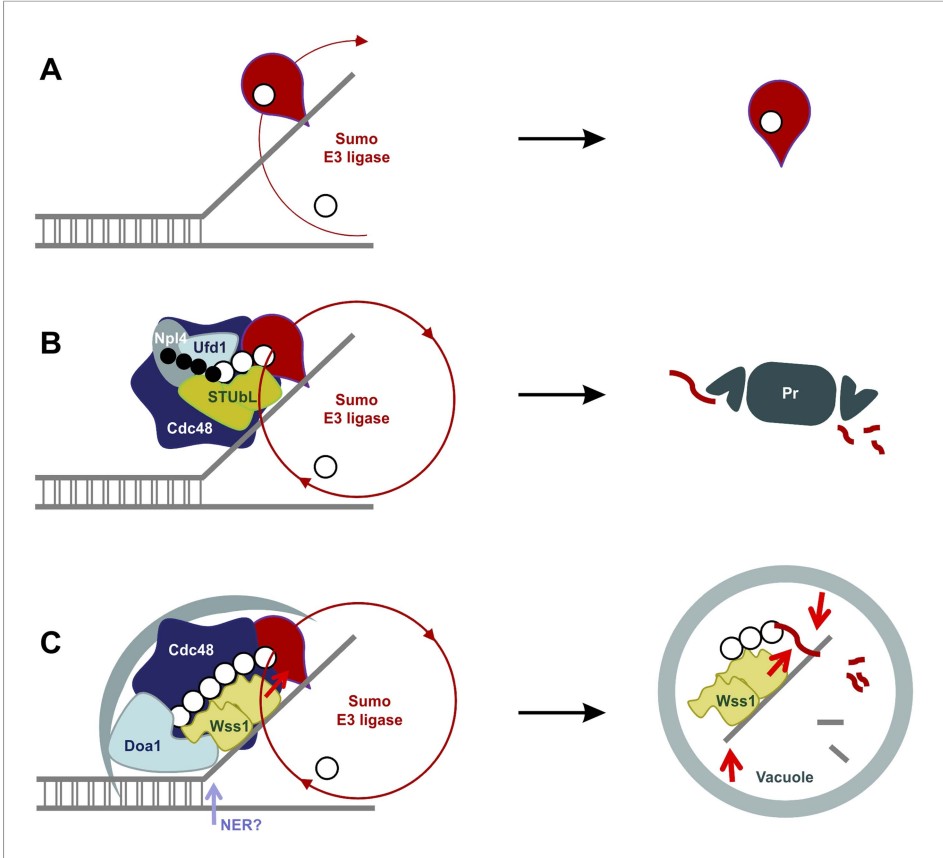

**Figure 11**. SUMO-dependent extraction of proteins from the chromatin. (**A**) ssDNA-activated SUMO E3 ligase sumoylates DNA-bound protein and induces its dissociation. (**B**) Delay in dissociation results in SUMO chain formation through multiple rounds of protein sumoylation. Subsequent ubiquitylation by STUbL promotes Cdc48/Npl4/Ufd1 loading, protein extraction and degradation via proteasome. (**C**) When the extraction is compromised (e.g., covalent protein–DNA adduct), the protein is processed by Cdc48/Wss1/Doa1 complex. Wss1 is targeted to sumoylated protein via its SIMs and promotes extension of SUMO chain that in return could further stimulate Wss1 accumulation and oligomerization at the site of DNA damage (Wss1 foci). Binding to ssDNA and oligomerization triggers metalloprotease activity of Wss1 and initiates substrate processing. The process is assisted by Cdc48 and Doa1 and finally ends in the vacuole.

of Wss1 with Cdc48 and Doa1 that exhibits specificity for SUMO, and promotes binding of HMW-SUMO to Cdc48, describes for the first time the SUMO ligase activity of Wss1, and demonstrates that Wss1 moves from nuclear foci to the vacuole during stress, reinforcing numerous intriguing links between SUMO-dependent DDR and vacuolar degradation. Finally, Wss1 proteolytic activity may be downregulated by auto-proteolysis to terminate the stress-activated proteolysis by Wss1.

## Materials and methods

### Yeast techniques and genetic manipulations

Strain construction, cell growth, yeast transformation, spotting assay followed standard protocols (*Guthrie and Fink, 2002*). Yeast knockout and TAP-tagged strains were obtained from Open Biosystems (GE Healthcare Dharmacon Inc., France). Other strains are listed in *Supplementary file 3*. Cell irradiation with UV light was performed with calibrated 254 nm UVC germicidal lamp. Cell treatment with 0–0.2 μg/ml 4-Nitroquinoline N-oxide (4-NQO, Sigma–Aldrich, France) was performed in log-phase cultures for 3 hr before analysis. For experiments with proteasome inhibitor MG132 cells were grown on synthetic medium containing L-proline instead of ammonium sulfate as the sole

nitrogen source and were permeabilized with sodium dodecyl sulfate (SDS, 0.003%) before addition of the inhibitor (75 µM final concentration for 3 hr) (*Liu et al., 2007*). Genotoxic treatment and colony-forming assays were performed as described previously (*Liu et al., 2002*).

## Recombinant proteins and mutagenesis

Plasmid vectors used for expression of recombinant proteins are listed in *Supplementary file 4*. Soluble HA-Wss1 proteins were produced in ArcticExpress (DE3) cells (Stratagene, Agilent Technologies, Santa Clara, CA, USA). Induction was performed in 1 l culture with 50 µM IPTG at 12°C overnight. Cells were collected, washed with PBS, resuspended in 10 ml of lysis buffer (LyB, 150 mM NaCl, 2 µM $ZnCl_2$, 1% Triton X-100, 50 mM Tris pH 7.4) supplemented with protease inhibitor cocktail w/o EDTA (Pierce, Thermo Fisher Scientific, France), and disrupted by sonication. All protein preparation and manipulation procedures were performed on ice or at 4°C, unless otherwise indicated. Cell lysate was cleared by centrifugation (25,000×g, 0.5 hr, Avanti J-26XP Beckman Coulter). The supernatant was filtered through sterile 0.2-µm filter and added to 200 µl of pre-equilibrated anti-HA affinity beads (Sigma–Aldrich, France). After incubation (with permanent rotation) for 1 hr in a cold room, beads were collected, washed three times with 2 ml LyB, and once with 2 ml of elution buffer (ElB, 150 mM NaCl, 0.1% Triton X-100, 50 mM Tris pH 7.4) supplemented with 10% glycerol. The HA-Wss1 proteins were eluted by 1 mg/ml HA peptide (Anaspec, Fremont, CA, USA) in ElB-10% glycerol, aliquoted, and stored at −80°C. For soluble MBP-Wss1 fusion proteins similar procedure was applied except that the expression was done in *BL21pLysS* (*DE3*) cells (Life Technologies, France), and induction was performed with 100 µM IPTG at 20°C overnight. The protein was purified on amylose resin (New England Biolabs, Ipswich, MA, USA) and eluted with 10 mM maltose in ElB-10% glycerol. For preparation of Wss1-his6 proteins from inclusion bodies, the expression was induced in 1 l of *BL21* (*DE3*) (Life Technologies) culture with 1 mM IPTG for 4 hr at 37°C. Cells were collected, washed with PBS, resuspended in 20 ml of suspension buffer (SuB, 50 mM NaCl, 1% Tx-100, 5 mM β-metrcaptoethanol and 20 mM HEPES, pH 7.5) supplemented with 1 mM PMSF and protease inhibitor cocktail w/o EDTA (Pierce) and disrupted by sonication. Protein inclusion bodies were collected by centrifugation (25,000×g, 0.5 hr) and washed with brief sonication in SuB two more times. The pellet was resuspended in 20 ml bind buffer (BiB, 7 M urea, 250 mM NaCl, 5 mM β-metrcaptoethanol, and 20 mM HEPES, pH 7.5) supplemented with 2% SDS (per liter of growth) and shaken at 37°C to solubilize the inclusion bodies. The protein containing solution was centrifuged at room temperature (25,000×g, 0.5 hr). The supernatant was diluted fourfold with BiB to reduce SDS concentration to 0.5%, loaded on Ni-NTA resin (2 ml Ni-NTA Superflow, Qiagen, France). The column was washed with 20 ml of BiB and the protein was eluted with BiB supplemented with 200 mM imidazole. The purified Wss1 was refolded by stepwise dialysis using 3 kD cut-off membrane (Spectra/Por, Spectrum Laboratories Inc., Piscataway, NJ, USA) by gradually decreasing the concentration of urea in BiB from 7 M to 1.5 M. The BiB was also supplemented with 2 mM $CaCl_2$, 5 µM $ZnCl_2$, and 0.1% Triton X-100. The final dialysis step was done in 150 mM NaCl, 0.1% Triton X-100, and 20 mM HEPES, pH 7.5 buffer with 10% glycerol. The Wss1 proteins were aliquoted and stored at −80°C. The recombinant HA-Wss1 protein and HA-Wss1-AQA mutant were prepared similarly except that the Ni-NTA purification step was omitted. When examining the effect of various additives on Wss1 refolding and activity, all molecules, except SDS (0.1% final concentration) were added directly into protein solution before dialysis: heparin (200 µg/ml sodium salt, Sigma–Aldrich), plasmid DNA (100 µg/ml pMAL-c2), and ssDNA (100 µg/ml M13mp18 single-stranded DNA, New England Biolabs). N-terminal protein sequencing was performed by automated Edman degradation (LF 3400; Beckman Instruments). All other recombinant proteins were prepared following previously published protocols (*Supplementary file 4*).

## SUMO-PA and Ub-PA synthesis

SUMO- and Ub-phosphoric acids were synthesized from Smt3(1–97)-MESNa and Ub(1–75)-MESNa, which were produced as previously reported (*Wilkinson et al., 2005*) and lyophilized. For reaction, 5 mg of MESNa thiol ester was solubilized in 0.5 M solution of aminomethylphosphonic acid solution (AMPA-Na, pH 8.5) containing 1 mM N-hydroxysuccinimide and incubated overnight at 37°C. The reaction was analyzed by following the loss of protein thiol ester with concomitant appearance of protein phosphonate by high-performance liquid chromatography (HPLC, Waters). After completion,

the reaction was dialyzed against 50 mM HCl, and lyophilized. The resulted SUMO-PA and Ub-PA were ~80% pure (HPLC) and were used without further purification.

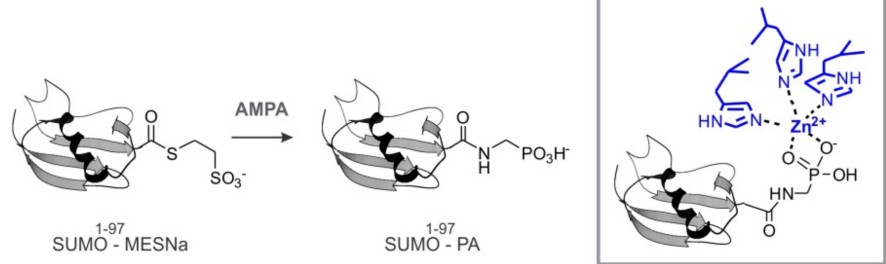

## Binding assays

Commercial affinity matrices anti-HA and anti-Myc affinity gels (Sigma–Aldrich), IgG- and glutathione-Sepharose (GE Healthcare Life Sciences, France), amylose resin (New England Biolabs) and Ni-NTA Superflow (Qiagen) were used for immunoprecipitation and pull-down assays. Other affinity beads were prepared with SUMO (PA), Ub (PA), and MBP-tagged recombinant proteins by using CNBr-activated Sepharose 4B (GE Healthcare Life Sciences) according to the manufacturer's protocol. The binding assays were performed in 150 mM NaCl, 50 mM Tris pH 7.4 and 0.1% or 1% (high stringency buffer) of Triton X-100. In vitro interaction assays were performed with 10–100 µg/ml of recombinant proteins for 1 hr with permanent rotation at 4°C. After incubation the beads were washed with the same buffer and bound proteins were eluted with 2× Laemmli buffer, unless otherwise indicated.

## In vitro sumoylation

In vitro sumoylation reactions were performed as described (*Johnson and Gupta, 2001*) in 150 mM NaCl, 10 mM $MgCl_2$, 0.1 mM DTT, 20 µg/ml bovine serum albumin (Fraction V, Sigma–Aldrich), and 20 mM HEPES (pH 7.5). The reaction mix contained 50 µg/ml SUMO or reductively methylated SUMO (prepared as described, *White and Rayment, 1993*), 2 µg/ml Uba2/Aos1, 10 µg/ml Ubc9, 4 mM ATP and either 1 µg/ml or 10 µg/ml of purified HA-Wss1. For auto- and Cdc48- sumoylation assays 10 µg/ml of HA-Wss1 and 20 µg/ml of Cdc48 and Doa1 was used. The reactions were conducted for 0.5–2 hr at 30°C. SUMO chains for Wss1 binding and proteolysis assays were prepared similarly with 100–200 µg/ml SUMO, 5 µg/ml Uba2/Aos1, 10 µg/ml Ubc9, 5 mM ATP, and 10 µg/ml of truncated Siz2 ligase (*Takahashi et al., 2003*) incubated at 30°C overnight and dialyzed against 150 mM NaCl, 50 mM Tris pH 7.4, and 0.05% of Triton X-100.

## In vitro proteolysis

In vitro proteolysis reactions were performed in 150 mM NaCl, 50 mM Tris pH 7.4, and 0.05% of Triton X-100. The reactions were conducted in a PCR block at 25°C. The reaction mix contained 50–200 µg/ml of refolded recombinant HA-Wss1 protein. To activate Wss1 protease, 0.5 mM thiram (dimethylcarbamothioylsulfanyl N,N-dimethylcarbamodithioate, 100 mM stock solution in DMSO), 1 mM APMA (4-aminophenylmercuric acetate, 200 mM stock solution in DMSO), and 2.5 µM DNA oligonucleotide (mbpTop1d, CGTCATCAGAAGACAACTCATGATTAACTTTGGAAGC ATCAGCAATAGTCGAGCTC GAATTAGTCTGCG, 100 µM stock solution in water). Wss1 substrates (SUMO chains, His6-Ub-SUMO-HA) were used at 100–200 µg/ml concentrations. Fluorescence anisotropy measurements with SUMO1-FP substrate were performed on a MOS-450 spectrometer (BioLogic, Inc.) in a 150 ml quartz cuvette at 25°C essentially as described (*Geurink et al., 2012*).

## Western blotting

The antibodies used for western blotting were anti- SUMO (Smt3 y-84, Santa-Cruz, Dallas, TX, USA), anti-Ub (clone 6C1, Sigma–Aldrich), anti-HA (clone 16B12, Covance, France), anti-MBP (clone MBP-17, Sigma–Aldrich), anti-Flag (FLAG M2, Sigma–Aldrich), anti-cMyc (Sigma–Aldrich), anti-Cdc48 (VCP antibody, Cell Signaling, Danvers, MA, USA), anti-GST (clone 8–326, Pierce), anti-TAP (Pierce),

anti-tubulin (Sigma), anti-histone H3 (ab46765, Abcam, France) secondary anti-rabbit HPR-coupled (Pierce), and secondary anti-mouse HPR-coupled (Pierce). Whole cell lysate (WCL) was prepared by heating yeast cells in WCL-buffer (100 mM NaOH, 2% SDS, 50 mM EDTA, and 2% β-mercaptoethanol) as described (*von der Haar, 2007*). The isolation of chromatin fraction was performed as described (*Verma et al., 2011*). The chromatin proteins were then solubilized by heating in WCL-buffer. Protein concentration was measured by Nanodrop and 80 µg of protein per lane of gel were generally used for western blot analysis. NuPAGE 3–8% Tris-acetate and 4–12% (and 4–20%) Bis-Tris precast polyacrylamide gels were used (Life Technologies).

## Isolation of STUbl substrates

To study the effect of Wss1 expression on STUbL substrates, SUMO conjugates were isolated from *EJY251-11b* cells containing *p315-PGAL-HFSMT3* plasmid (*Johnson et al., 1997*). Cells were transformed with HA-Wss1 constructs (pYEPGAP-URA3) and grown at 30°C in selective media containing L-proline as the sole nitrogen source and 2% galactose/1% raffinose as a carbon source. Cells were permeabilized with 0.003% SDS before treatment with 75 µM MG132 for 3 hr. Cells were harvested and lysed in hot WCL-buffer as described above. The WCLs were extracted by methanol-chloroform-water (4:1:3), proteins were precipitated with methanol and briefly dried (*von der Haar, 2007*). The resulted pellet was solubilized in load buffer (LoB, 6 M urea, 10 mM NaCl, 50 mM Tris, pH 7.4), loaded onto Ni-NTA Superflow beads, rinsed with LoB, and eluted in LoB supplemented with 200 mM imidazole or directly by 2× Laemmli buffer. The eluted proteins were analyzed by western blotting.

## Mass spectrometry

To identify Wss1 partner proteins, we used a strain expressing Wss1-Myc$_{13}$ from endogenous loci. Cell lysates were prepared from 2 l of log-phase *WSS1-MYC13* culture and parental strain as a control. Cells were collected, frozen at −80°C, and cell lysate was prepared essentially as described (*Puig et al., 2001*) except that the dialysis step was omitted. The lysate was incubated with 250 µl of anti-c-Myc agarose affinity gel (Sigma–Aldrich) for 1 hr at 4°C; the gel was washed with cell lysis buffer (CLB), and the bound proteins were eluted with 0.1 M ammonium hydroxide and lyophilized. For MS analysis the samples were solubilized in 1× Laemmli buffer and separated by gel electrophoresis. To identify Doa1 partner proteins, cells lysates were prepared from 4 l of log-phase *BY4742* culture and pre-cleared by passing through 2 ml of Ni-NTA Superflow gel (Qiagen). To one half of the lysate 1 mg of purified recombinant HF-Doa1 was added, and the mixture was passed through 1 ml of Ni-NTA Superflow gel. Another half of the lysate was treated the same way but without HF-Doa1 and serves a control. The bound proteins were eluted with 1× Laemmli buffer and were cleaned by running a very short 9% SDS gel (~3 mm long) followed by staining and extensive wash. In all samples the gel was excised into slices and digested by trypsin (*Shevchenko et al., 1996*). The resulting peptides were dissolved in a loading buffer (6% acetic acid, 0.005% heptafluorobutyric acid and 5% acetonitrile) and analyzed by reverse phase liquid chromatography coupled with tandem MS (LC-MS/MS) (*Peng and Gygi, 2001*) on an LTQ-Orbitrap hybrid mass spectrometer (Thermo Electron, San Jose, CA, USA).

The analysis of all MS/MS spectra was performed using the SEQUEST algorithm (version 27) (*Eng et al., 1994*) and a composite target/decoy database. While the target proteins were derived from *Saccharomyces cerevisiae* sequences and known contaminant proteins, such as porcine trypsin and human keratins, the decoy components contained the randomized sequences of all target proteins (*Peng et al., 2003*). The database search was processed without enzyme restriction and with mass tolerance of 3.05 Da for precursor ion. Modifications were permitted to allow for the detection of the following (mass shift shown in Daltons) oxidized methionine (+15.9949 Da) and acrylamide-cysteine adduct (+71.0371 Da). The results of SEQUEST were filtered according to XCorr and Cn to obtain a false-positive rate of 1% in peptide identification.

## RMN

The Wss1 VIM (209–219) peptide was synthesized in Anaspec. RMN data were acquired at 278 K on an Agilent VNMRS 800 MHz spectrometer equipped with a triple-resonance HCN cold probe and pulsed field gradients. The 1H frequencies assignment was achieved with the combined use of 1H-1H- DQF-COSY, TOCSY (mixing time 80 ms) and NOESY (mixing time: 250 ms) spectra. CYANA 2.1

was used to calculate the structure of Wss1-VIM peptide (*Guntert et al., 1997*), using NOE restraints measured from the 1H-1H NOESY spectrum. From the observation of NOE cross peaks characteristic of alpha helical conformation daN(i,i+2) daN(i,i+3) and dNN(i,i+2) (*Wuthrich, 1986*) and the chemical shift analysis of the Ha chemical shifts of the peptide using the program TALOS (*Shen et al., 2009*) showing helical propensities for all residues, we applied PHI and PSI dihedral angles restraints corresponding to canonical helix values along the peptide backbone. The automatically assigned NOEs were calibrated within CYANA according to their intensities. After seven rounds of calculation (10,000 steps per round), 120 cross-peak NOE assignments were used in the final calculation. The 10 lowest energy conformations of the peptide have no constraint violations and show a backbone root-mean-square deviation of $0.2 \pm 0.1$ Å and a heavy atom root-mean-square deviation of $0.8 \pm 0.2$ Å.

## Bioinformatics, molecular docking, and modeling

Conserved regions in WLM proteins were identified using BLOCKS Database looking for the proteins documented in the Prosite Database (http://blocks.fhcrc.org/help). Consensus secondary structure prediction was performed on SYMPRED web server (http://www.ibi.vu.nl/programs/sympredwww/). 3D structure prediction and modeling of WLM and SprT proteins were performed using the Phyre2 server (*Kelley and Sternberg, 2009*; http://www.sbg.bio.ic.ac.uk/phyre2/html/page.cgi?id=index). The PatchDock web server was used to perform the docking runs (*Schneidman-Duhovny et al., 2005*; http://bioinfo3d.cs.tau.ac.il/PatchDock/). Protein interaction network (medium confidence) was constructed using STRING (Search Tool for the Retrieval of Interacting Genes/Proteins: http://string-db.org/), and analyzed in GO terms with Cytoscape_v2.8.3.

## Isolation of Wss1 complexes and pull-down assays

To isolate Wss1-interacting proteins, lysates were prepared from cells transformed with HA-Wss1 constructs treated or not with 0.2 µg/ml of 4-NQO. Cells from an A600 $\cong$ 15 culture were harvested by centrifugation, washed in water, and processed by glass bead beating (Disruptor Genie, Scientific Industries) for 5 min at 4°C in 350 µl CLB (150 mM NaCl, 50 mM Tris pH 7.4, 0.5% Triton X-100) supplemented with 30 mM NEM and protease inhibitor cocktail w/o EDTA. The sample was centrifuged at 20,000×g for 15 min at 4°C, the supernatant was collected and the pellet was extracted two more times as described above. The supernatant fractions were pooled and added to 50 ml of anti-HA beads pre-equilibrated with CLB. The beads were rotated for 1 hr at 4°C, washed 3× 500 µl CLB, and bond proteins were eluted either with 100 µl of 2× Laemmli buffer w/o reducing agent (denaturing conditions) or with 100 µl of 1 mg/ml HA peptide in CLB (native complexes). To study Wss1 proteolysis the native complexes were incubated in CLB either with or without a protease inhibitor (1 mM PMSF, 2 mM OPA, or 30 µM MG132), and the reaction was quenched with 2× Laemmli buffer and analyzed by western blotting. Similar protocol was used for pull-down assays with CNBr-crosslinked MBP-Wss1 beads and IgG beads.

Elution of protein complexes from IgG beads by cleavage of TAP-tag with TEV protease was performed in CLB buffer supplemented with 0.5 mM EDTA and 1 mM DTT (*Puig et al., 2001*). Recombinant TEV protease was added to the beads suspension (50 µg/ml final concentration), and the beads were rotated 2 hr at 20°C. The eluate was recovered after centrifugation and analyzed by western blotting.

## Microscopy and cytofluorimetry

Standard live yeast cell microscopy techniques were used (*Guthrie and Fink, 2002*). Cells were grown to log-phase on glucose- or galactose-containing medium before harvesting. For staining of vacuolar membranes, cells were incubated for 0.5 hr with 1 µg/ml FM4-64 (Life Technologies) with shaking at 30°C. Cells then were resuspended in fresh medium lacking the dye and were allowed to grow for 1 hr at 30°C. For DNA staining, cells were incubated for 0.5 hr with 2.5 µg/ml of DAPI (Sigma–Aldrich) before observation. The expression of Wss1-GFP under control of inducible MET25 promoter (pUG35 and pUG36 vectors) was regulated by varying methionine concentration in the growth medium from 10 mg/l (low level of expression) to 0 mg/l (high level of expression). Cells were analyzed in fresh medium on concanavalin A (Sigma–Aldrich)-coated slides using AxioImager Z1 Zeiss microscope. Live cell cytofluorimetry was performed with Becton Dickinson LSR II cytofluorimeter.

## Acknowledgements

We thank S Biggins, ES Johnson, T Chernova, S Jentsch, X Zhao, GPH van Heusden, M Hochstrasser, and WT Wickner for providing yeast strains and expression vectors, H Ovaa and P Geurink for SUMO-FP reagents, G Curien for help with fluorescence anisotropy measurements, N Degtyareva and all members of BIOMICS and Wilkinson laboratories for various reagents, procedures, and helpful discussions. This work was supported by grants NIH GM030308 and GM09329 to KDW.

## Additional information

### Funding

| Funder | Grant reference | Author |
|---|---|---|
| National Institute of General Medical Sciences (NIGMS) | GM030308 | Keith D Wilkinson |
| National Institute of General Medical Sciences (NIGMS) | GM09329 | Keith D Wilkinson |

The funder had no role in study design, data collection and interpretation, or the decision to submit the work for publication.

### Author contributions

MYB, Conception and design, Acquisition of data, Analysis and interpretation of data, Drafting or revising the article; JEM, AF, ES, DFL, Acquisition of data, Analysis and interpretation of data; NA, AVR, Conducted new studies on cysteine switch, Acquisition of data, Analysis and interpretation of data; XG, KDW, Analysis and interpretation of data, Drafting or revising the article

## Additional files

### Supplementary files

• Supplementary file 1. Wss1-interacting proteins identified by mass spectrometry.

• Supplementary file 2. Doa1-interacting proteins identified by mass spectrometry.

• Supplementary file 3. Yeast strains used in this study.

• Supplementary file 4. Plasmids used in this study.

• Supplementary file 5. Model of the WLM domain of Wss1. The structure of the WLM domain of Wss1 was modeled using the Phyre2 server.

• Supplementary file 6. Model of the SprT domain of DVC1/Spartan. The structure of the SprT domain of DVC1/Spartan was modeled using Phyre2 server.

• Supplementary file 7. Structure of abylysin (4JIU). This is a minigluzincin family metalloprotease used for modeling the WLM and SprT structures.

• Supplementary file 8. Model of the Wss1 VIM peptide docked to Cdc48. The yeast N-Cdc48 domain (residues 23–186, magenta) was superimposed on 3TIW, the human P97 structure (residues 23–186, cyan) bound to the gp78 VIM (blue) using the align command in PyMOL. The PatchDock web server (*Schneidman-Duhovny et al., 2005*) was then used to perform the docking of the Wss1 VIM NMR structure (red) determined above (*Figure 5—figure supplement 3A*) to Cdc48. This is a pyMOL .pse file.

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
