## [Decision Letter]

Thank you for sending your work entitled “Wss1 metalloprotease partners with Cdc48/Doa1 in processing genotoxic SUMO conjugates” for consideration at *eLife*. Your article has been favorably evaluated by Randy Schekman (Senior Editor) and two reviewers, one of whom is a member of our Board of Reviewing Editors.

The Reviewing Editor and the other reviewer discussed their comments before we reached this decision, and the Reviewing Editor has assembled the following comments to help you prepare a revised submission.

The manuscript extends the recent work by Stingele et al. in analyzing the function of Wss1 during DNA damage and repair events. Both by in vivo and in vitro experiments show that Wss1 promotes SUMOylation and is also responsible for clearing high molecular weight SUMO conjugates by forming a ternary complex with Cdc48 and its adaptor Doa1. The study systematically characterized the SUMO binding by Wss1 in an elegant way especially analyzing the contribution of WLM domain towards SUMO binding. Wss1 can bind to poly SUMO conjugates and also act as a SUMO ligase to promote SUMO chain formation. Proteolytic activity of Wss1 was tested in vivo and once again they showed that the metalloprotease activity is indeed SUMO-dependent during genotoxic stress. In addition, Wss1 is localized in vacuoles upon genotoxic stress in a SUMO-dependent manner indicating to the link between SUMO-dependent DDR and autophagic events.

There are three major points that should be addressed before the manuscript is acceptable for publication:

1) The analysis of the protease activity. The authors mention reported isopeptidase activity of Wss1 towards Ub-SUMO linkages, and state that they cannot see isopeptidase activity. However, the authors test the activity of Wss1 in vitro only towards a peptide-linked fusion construct (i.e. not an isopeptide). These experiments were also done in absence of DNA, which they show activates the protease. The fact that the WLM catalytic domain contributes to SUMO binding, recognizes the C-terminus of SUMO, and binds very nicely to a novel activity-based SUMO-PA probe suggest that Wss1 may be a deSUMOylase. This is somewhat suggested by cell data (Figures 5 and 6). In principle, testing deSUMOylase activity (against an isopeptide bond) should be a straightforward experiment that may put to rest some of the apparent issues in the literature. For this SUMO-FP reagents could be utilized.

2) Wss1 as a SUMO ligase. The authors surprisingly show that Wss1 contains E3 ligase activity, and this depends on functional SIMs in Wss1. By over expressing various Wss1 mutants and interacting partners Wss1 contributes to their SUMOylation. Their model seems to be that the two SIMs act in an E3-like fashion. If this was the case, why is the C-terminal fragment of Wss1 inactive in ligase assays? Also, why are other proteins with multiple SIMs, such as RNF4, not also SUMO ligases? It would be very satisfying to see additional residues in Wss1 be involved in ligase activity.

3) The authors claim that Wss1 is involved in vacuolar degradation during DDR events. The authors should provide more direct evidence about the link between Wss1 and vacuolar degradation, e.g. the authors should show the co-localization of Wss1 and Doa1 or Cdc48 in the presence of 4-NQO. Secondly, they should test if any other component of the DDR is found in the vacuole in the presence or absence of Wss1.

[Editors' note: further revisions were requested prior to acceptance, as described below.]

Thank you for resubmitting your work entitled “Wss1 metalloprotease partners with Cdc48/Doa1 in processing genotoxic SUMO conjugates” for further consideration at *eLife*. Your revised article has been favorably evaluated by Randy Schekman (Senior Editor), a Reviewing Editor, and one reviewer. The manuscript has been improved but there are some remaining issues that need to be addressed before acceptance, as outlined below:

1) Readers may not be familiar with the structure and mechanism of thiram, which should be discussed in more detail.

2) The authors leave the reader with a discussion of a protease activity towards SUMO that is not a deSUMOylase (isopeptidase) activity, yet still ‘processes’ SUMO chains, based on gel analysis. It should be mentioned somewhere that the mechanism, site and products of processing are unclear.

3) Cdc48 is a hexamer. It is unclear what the authors mean by a 1:1:1 complex – please clarify.

---

## [Author Response]

*1) The analysis of the protease activity. The authors mention reported isopeptidase activity of Wss1 towards Ub-SUMO linkages, and state that they cannot see isopeptidase activity. However, the authors test the activity of Wss1 in vitro only towards a peptide-linked fusion construct (i.e. not an isopeptide). These experiments were also done in absence of DNA, which they show activates the protease. The fact that the WLM catalytic domain contributes to SUMO binding, recognizes the C-terminus of SUMO, and binds very nicely to a novel activity-based SUMO-PA probe suggest that Wss1 may be a deSUMOylase. This is somewhat suggested by cell data (*Figures 5 and 6*). In principle, testing deSUMOylase activity (against an isopeptide bond) should be a straightforward experiment that may put to rest some of the apparent issues in the literature. For this SUMO-FP reagents could be utilized*.

We performed the experiments proposed by the reviewers.

First of all, we established the conditions to produce active recombinant Wss1 metalloprotease. This led us to the discovery of cysteine-regulation (cysteine switch) mechanism and the fact that DNA is not required for Wss1 activity (Figure 2 and Figure 2—figure supplement 1, Figure 2—figure supplement 2 and Figure 2—figure supplement 3). In these experiments, self-cleavage of the protease is a measure of the activation of Wss1 protease.

We too were surprised that isopeptidase activity was not demonstrable with the activated Wss1 protease. Binding to SUMO-PA may simply mean that the metal site is nearby to the SUMO binding site, not necessarily that it is catalytically involved as an isopeptidase.

We examined the active Wss1 for cleavage of three different SUMO substrates: polySUMO-chains (multiple iso-peptide bonds), SUMO1-FP substrate (monoSUMO fluorescent iso-peptide substrate) and linear His6-Ub-SUMO-HA (peptide-linked fusion construct). The proteolysis by Wss1 was compared with SUMO-isopeptidase Ulp1, and Ub-isopeptidase Usp2 (new Figure 3 and Figure 3—figure supplement 1). Our data suggest that Wss1 is not a SUMO-isopeptidase but rather processes polySUMO chains, probably due to metalloprotease activity in a hinge region of the polySUMO conjugates.

*2) Wss1 as a SUMO ligase. The authors surprisingly show that Wss1 contains E3 ligase activity, and this depends on functional SIMs in Wss1. By over expressing various Wss1 mutants and interacting partners Wss1 contributes to their SUMOylation. Their model seems to be that the two SIMs act in an E3-like fashion. If this was the case, why is the C-terminal fragment of Wss1 inactive in ligase assays? Also, why are other proteins with multiple SIMs, such as RNF4, not also SUMO ligases? It would be very satisfying to see additional residues in Wss1 be involved in ligase activity*.

A priori, the presence of multiple properly oriented SIMs may be sufficient to confer SUMO ligase-like activity to a protein. For Slx5, the yeast RNF4 homolog, such activity has also been reported (22), even though the physiological relevance of this finding is unclear. Interestingly, among other known non-RING SUMO E3-ligases – including RanBP2, polycomb Pc2, topors, and Rad18 – all, except RanBP2, catalyze sumoylation by SIM-dependent mechanisms (37; 50; 53). We discuss this now in the revised manuscript.

Nonetheless, the C-terminal fragment of Wss1 was indeed inactive in ligase assays. Our data and the data of [66] suggest that Wss1 protein is prone to oligomerization and this may be important for regulation of the protein ligase function. Although we didn’t address this question in detail, it seems plausible that, like for some other metalloproteases, the protease domain itself oligomerizes. Oligomerization increases the number of SUMO-binding sites and hence may further promote SUMO ligase activity (new Figure 4—figure supplement 2). Corroborating this conclusion we found that induction of Wss1 oligomerization by OPA treatment stimulates its auto-sumoylation (new Figure 4). Finally, It may be that the SUMO binding by WLM is necessary to form a functional catalytic site, especially since SIMs present such a small interacting surface and may not be able to properly orient the SUMO for optimal catalysis.

*3) The authors claim that Wss1 is involved in vacuolar degradation during DDR events. The authors should provide more direct evidence about the link between Wss1 and vacuolar degradation, e.g. the authors should show the co-localization of Wss1 and Doa1 or Cdc48 in the presence of 4-NQO. Secondly, they should test if any other component of the DDR is found in the vacuole in the presence or absence of Wss1*.

Because of time and space constraints we were unable to complete the requested studies. We agree with the reviewers that this point is not developed (and we emphasize this in the text). However, we feel that the questions asked by the reviewers are important enough that they should be addressed in depth in a dedicated independent study, rather than appended to this manuscript.

We think it is important to include these preliminary findings because they point to an entirely new aspect of Wss1 function that could have profound importance. Our observations are that: (1) Three different genotoxic stresses (UV, *smt3-331*, tdpΔ) all resulted in accumulation of Wss1 in the vacuole; (2) A third of the highest-ranked proteins associated with Wss1 were proteins associated with vesicle transport and vacuole; (3) The major Wss1 interactor, a Cdc48/cofactor Doa1 complex, is unequivocally involved in regulation of two main vacuolar pathways, autophagy and vesicle sorting. This demonstrates that there is a significant connection between Wss1 function and this organelle, seemingly in response to DDR and other stressors.

We feel there is little mechanistic detail to be gained from the simple co-localization studies requested. It is already known that DDR induces autophagy and DDR components are degraded by the vacuole/lysosome (11; 59; 73), and that sumoylated Cdc48 is targeted to the vacuole (35). Thus, it may not be surprising that Wss1 complexes with Doa1 and Cdc48 would co-localize with substrates in the vacuole.

[Editors' note: further revisions were requested prior to acceptance, as described below.]

*1) Readers may not be familiar with the structure and mechanism of thiram, which should be discussed in more detail*.

We added one phrase in the Results explaining that Th modifies proteins by thiol-disulfide exchange, as well as two supporting new references. We also added new Figure 2—figure supplement 1, which illustrates this mechanism.

*2) The authors leave the reader with a discussion of a protease activity towards SUMO that is not a deSUMOylase (isopeptidase) activity, yet still ‘processes’ SUMO chains, based on gel analysis. It should be mentioned somewhere that the mechanism, site and products of processing are unclear*.

We have added several phrases in the Discussion outlining that, although the exact mechanism of how Wss1 processes SUMO conjugates remains unclear, one possible scenario may be the activation of metalloprotease and cleavage within a hinge region of the polysumoylated proteins. Alternatively, our finding that Wss1 binds to SUMO-PA, self-cleaves at C-terminus and produces a ladder of proteolytic fragments upon activation by ssDNA may suggest that it functions as a SUMO-targeted carboxypeptidase.

*3) Cdc48 is a hexamer. It is unclear what the authors mean by a 1:1:1 complex - please clarify*.

Actually, that is why we inserted the pictograms of Cdc48 complexes Figure 2—figure supplement 1, where it is clear that 1:1:1 complex means 6:6:6 or one monomer of Wss1 per monomer of Cdc48.